1 # Source attribution of near-surface ozone trends in the United States during 1995–2019

Pengwei Li[1], Yang Yang[1*], Hailong Wang[2], Su Li[1], Ke Li[1], Pinya Wang[1], Baojie
Li[1], Hong Liao[1]
[1]Jiangsu Key Laboratory of Atmospheric Environment Monitoring and
Pollution Control, Jiangsu Collaborative Innovation Center of Atmospheric
Environment and Equipment Technology, School of Environmental Science
and Engineering, Nanjing University of Information Science and Technology,
Nanjing, Jiangsu, China
[2]Atmospheric Sciences and Global Change Division, Pacific Northwest
National Laboratory, Richland, Washington, USA
*Correspondence to yang.yang@nuist.edu.cn

**Abstract**

Emissions of ozone ($O_3$) precursors in the United States have decreased in recent decades, and near-surface $O_3$ concentrations showed a significant decrease in summer but an increase in winter. In this study, an $O_3$ source tagging technique is utilized in a chemistry-climate model to investigate the source contributions to $O_3$ mixing ratios in the U.S. from various emitting sectors and regions of nitrogen oxides ($NO_x$) and reactive carbon species during 1995–2019. We show that domestic emission reductions from energy and surface transportation are primarily responsible for the decrease in summertime $O_3$ during 1995–2019. However, in winter, the emission control also weakens the $NO_x$ titration process, resulting in considerable increases in $O_3$ levels from natural sources. Additionally, increases in aviation and shipping emissions and transpacific transport of $O_3$ from Asia largely contribute to the winter $O_3$ increase. We also found that changes in large-scale circulation favoring $O_3$ transport from upper atmosphere and foreign transport from Asia also explain 15% of the increase in the U.S. near-surface $O_3$ levels in winter.

## 1. Introduction

Ozone ($O_3$) near the surface has a significant impact on air quality and public health (Haagen-Smit, 1952; Fleming et al., 2018). Since the increase in anthropogenic emissions of $O_3$ precursors from preindustrial times, $O_3$ has now become the third most important anthropogenic greenhouse gas in the troposphere (Myhre et al., 2013). Major sources of $O_3$ in the troposphere include the transport from the stratosphere and formation through photochemical reactions within the troposphere involving two chemically distinct groups of precursors: nitrogen oxides ($NO_x$) and reactive carbon species, including carbon monoxide (CO), methane ($CH_4$), and non-methane volatile organic compounds (NMVOCs) (Atkinson, 2000). $O_3$ precursors come from a variety of sectors, and its relatively long lifetime of about 22 days (Stevenson et al. 2006) favors the long-range transport of $O_3$. Due to the nonlinearity of the $O_3$ production and its associated dependence on precursor emissions (Seinfeld and Pandis, 2006), attributing $O_3$ pollution to its sources is complicated.

Since the 1980s, $O_3$ precursor emissions have significantly reduced in the United States (Duncan et al., 2016; Xing et al., 2013; Zhang et al., 2016; Zhang et al., 2021). However, due to the nonlinear production chemistry of $O_3$, complex seasonal meteorological influence, and long-range transport from foreign source regions, domestic emissions reductions do not imply a decrease in seasonal and annual $O_3$ concentrations. According to remote surface measurements (Cooper et al., 2020) and aircraft observations (Gaudel et al., 2020), the Sixth Assessment Report of the Intergovernmental Panel on Climate Change (Szopa et al., 2021) showed a decreasing trend in annual mean $O_3$ concentrations in the western U.S. but an increasing trend in the eastern U.S. since the mid-1990s. On the seasonal timescale, surface observations and modeling results showed that $O_3$ concentrations over the U.S. had decreased

in summer due to the reductions in domestic anthropogenic emissions and
increased in winter related to the weakened $NO_x$ titration since the late 1980s
(Cooper et al., 2012; Lin et al., 2017). It also shows that the increased
background $O_3$, especially due to an increased transport from Asia, can partly
offset the benefit of domestic emissions control over the western U.S. in
summer.
Source apportionment is a useful method for quantifying contributions to
air pollutants from specific source regions and/or sectors, which is beneficial to
emission control strategies (Yang et al., 2018). One method of obtaining an $O_3$
source-receptor relationship is to zero out or perturb emissions from a given
source region or sector in sensitivity simulations along with a baseline
simulation, which gives information about the response of $O_3$ to changes in
precursor emissions (e.g., Fiore et al., 2009; Hoor et al., 2009). However,
emission perturbation method requires many additional model simulations
when being used to estimate the impacts of changes in multiple sources (Koo
et al., 2009; Wang et al., 2014). The perturbation method may invalidate the
assumption of a linear relationship between the magnitude of the emission
perturbation and the magnitude of the $O_3$ change considering the nonlinearity
in $O_3$ chemistry, especially if large perturbations (e.g. zeroing out regional or
sector-wide emissions) are used. The tagging approach produces information
about the contribution of precursor emissions to the total amount of $O_3$ (Butler
et al., 2020). The perturbation and tagging methods are two different methods
answering different scientific questions, with the first for the impacts and the
last for the contributions (Grewe et al. 2010, Emmons et al. 2012, Clappier et
al. 2017 and Thunis et al., 2019). Both of these two methods can be used for
specific purpose to provide a comprehensive understanding of source-receptor
relationships between precursor emissions and $O_3$ concentrations.
The source tagging method has been widely adopted in regional air quality
models to examine the $O_3$ attribution in the U.S., China, and/or Europe (Gao et
al., 2016; Collet et al., 2018; Lupaşcu and Butler, 2019). In some regional
models, $O_3$ apportionment is based on the ratio of chemical indicators to
determine the regime of $O_3$ generation (e.g., VOC-limited or $NO_x$-limited
regimes) and then attribute the generation of $O_3$ to the tag carried by a certain
precursor (VOCs or $NO_x$), which however cannot simultaneously attribute $O_3$
production to $NO_x$ and VOCs, respectively (Dunker et al., 2002; Kwok et al.,
2015), while some models do not use the chemical indicators (Lupaşcu and
Butler, 2019; Mertens et al., 2020). In addition, due to the limitation in domain
size of regional air quality models, they are difficult to account for contributions
of intercontinental transport from several sources outside the model domain.
Recently, $O_3$ tagging techniques have been implemented in the global models
(e.g., Sudo and Akimoto, et al., 2007; Zhang et al., 2008; Emmons et al., 2012;
Grewe et al. 2017; Butler et al., 2018; Han et al., 2018; Bates and Jacob, 2020).
However, in many global models, $O_3$ is tagged by the production regions rather
than the precursor emission regions, so that $O_3$ can only be attributed to the
area where $O_3$ is generated, rather than the source of precursor emissions.
Here, based on a state-of-the-art tagging system implementation in a
global chemistry–climate model, the trends of near-surface $O_3$ mixing ratios in
the U.S. during 1995–2019 and the source attributions of the $O_3$ variations to
various emission sectors and regions of $NO_x$ and reactive carbon species are
investigated in this study. Mechanisms of explaining the $O_3$ trends that involve
changes in anthropogenic emissions and large-scale circulations are also
explored.
**2. Methods**
**2.1 Model Description**
Tropospheric $O_3$ mixing ratios are simulated using the Community
Atmosphere Model version 4 with Chemistry (CAM4-chem) (Lamarque et al.,
2012; Tilmes et al., 2015), which is the atmospheric chemistry component of
the Community Earth System Model (CESM), at a horizontal resolution of 1.9°
latitude by 2.5° longitude with 26 vertical levels extending to 40 km above the
surface. The height of bottom layer is about 120 m and there are about 4 layers
under 2 km. The model configuration uses a comprehensive tropospheric
chemistry mechanism based on the Model for Ozone and Related chemical
Tracers version 4 (MOZART-4) (Emmons et al., 2010, 2012). Model
configurations simulate wet deposition of gas species using the Neu and
Prather (2011) scheme. Dry deposition is represented following the resistance
approach originally described in Wesely (1989). Stratosphere-troposphere
exchange of $O_3$ is treated by setting $O_3$ to stratospheric values as their
climatological means over 1996–2005 at the tropopause (Lamarque et al.,
2012), which is affected by atmospheric circulation and experiences the same
loss rates as $O_3$ in the troposphere (Tilmes et al., 2016). Sea surface
temperatures and sea ice concentrations in our simulations are prescribed at
present-day climatological conditions. The zonal and meridional wind fields are
nudged towards the MERRA-2 (Modern Era Retrospective-Analysis for
Research and Applications Version 2) reanalysis (Gelaro et al., 2017) at a 6-
hourly relaxation timescale in this study to better constrain large-scale
circulations by observations. The CAM4-chem performance in simulating
tropospheric $O_3$ and precursors has been fully evaluated in Tilmes et al. (2015).
**2.2 Ozone Source Tagging Technique**
The novel $O_3$ source tagging technique implemented in the model was
developed by Butler et al. (2018), which can provide a separate source
apportionment of tropospheric $O_3$ to the two distinct groups of precursor
emissions, i.e., $NO_x$ and reactive carbon (CO, $CH_4$ and NMVOCs). The portion
of tropospheric $O_3$ that is attributable to the stratosphere-troposphere exchange
can also be quantified using this unique tagging technique. The source
attribution of $O_3$ requires two separate model runs with the tagging applied to
$NO_x$ and reactive carbon species, respectively. Details of the $O_3$ tagging
technique are described in Butler et al. (2018).

In this study, near-surface $O_3$ is attributed to emission sectors and regions.

Emissions from individual sectors, including agriculture (AGR), energy (ENE),
industry (IND), residential, commercial and other (RCO), surface transportation
(TRA), waste management (WST), international shipping (SHP) and biomass
burning (BMB) emissions, as well as chemical production in the stratosphere
(STR) and extra chemical production (XTR, a small amount of $O_3$ produced due
to the self-reaction of OH radicals and the reactions of $HO_2$ with certain organic
peroxy radicals) are tagged for both $NO_x$ and reactive carbon species. Aircraft
(AIR), soil (SOIL) and lightning (LGT) sources are separately tagged for $NO_x$
emissions, while solvents (SLV) and biogenic (BIO) sources are separately
tagged for NMVOCs emissions.

For the regional source attribution, we separately tag anthropogenic

sources from Africa (AFR), Central America (CAM), Europe (EUR), Middle East
(MDE), North America (NAM), East Asia (EAS), South Asia (SAS), Southeast
Asia (SEA) and rest of the world (ROW) (see Fig. 1 for the region map) and
natural sources (BMB, SOIL, LGT, BIO, STR and XTR). Additional tags for
methane ($CH_4$) and carbon monoxide (CO) are applied in both of the reactive
carbon tagging simulations that are used to attribute $O_3$ to emission sectors and
regions. We do not tag $CH_4$ by individual sources and the contributions of $CH_4$
from various sources are lumped in this study. It is because $CH_4$ has a relative
long lifetime in the troposphere and it is well mixed in the troposphere due to its
exceptionally low reactivity, which can contribute to $O_3$ formation at any location
in the troposphere where photochemical conditions are favorable (Fiore et al.,
2008). CO also has a longer lifetime and lower reactivity than most NMVOCs.
The lumped CO is tagged in the simulations for emission sectors, but not
specifically tagged in the simulations for emission regions due to the
computational limit.
**2.3 Emissions and Observation**

The global anthropogenic emissions, including $NO_x$, CO, NMVOCs, $SO_2$,

and $NH_3$, over 1990–2019 are from the Community Emissions Data System
(CEDS) version 20210205 (Hoesly et al., 2018) (See Table S1 and Figs. S1–
S3). Biomass burning emissions are obtained from the CMIP6 (Coupled Model
Intercomparison Project Phase 6) over 1990–2014 (van Marle et al., 2017) and
the emissions for the following five years (2015–2019) are interpolated from the
SSP2-4.5 forcing scenario (O'Neill et al., 2016). $NO_x$ emitted from soils and
biogenic NMVOCs from vegetation are prescribed as in Tilmes et al. (2015) and
are kept at the present-day (2000) climatological levels during simulations.
Lightning emissions of $NO_x$ are estimated online using the parameterization
based on simulated cloud top heights from Price et al. (1997), which is scaled
to provide a global annual emission of 3–5 Tg N $yr^{-1}$ (Lamarque et. al., 2012).
$CH_4$ is fixed at a global average level of 1760 parts per billion (ppb, volume ratio
in this study) during simulations.

Many studies have reported that the previous CEDS version 20160726

(hereafter $CEDS_{2016}$) has large biases in the regional emission estimates (e.g.,
Cheng et al., 2021; Fan et al., 2018). In this study, the CEDS version 20210205
is used (hereafter $CEDS_{2021}$), which builds on the extension of the CEDS
system described in McDuffie et al. (2020) and extends the anthropogenic
emissions to year 2019. It updates country-level emission inventories for North
America, Europe and China and has considered the significant emission
reductions in China since the clean air actions in recent years. The global total
$NO_x$ emission from $CEDS_{2021}$ is lower than that of $CEDS_{2016}$ after 2006 and it
shows a fast decline since then. In 2014, the global total anthropogenic
emission of $NO_x$ in $CEDS_{2021}$ is about 10% lower than the $CEDS_{2016}$ estimate.
This difference is mainly reflected in the $NO_x$ emissions in China and India.
$CEDS_{2021}$ has a lower estimate of the global NMVOCs emission than $CEDS_{2016}$
by more than 10% during the recent decades, attributed to lower emissions
from Africa, Central and South America, the Middle East and India. The using
of the $CEDS_{2021}$ emission inventory in this study could reduce the contributions
of $NO_x$ emissions from East Asia and South Asia to the U.S. $O_3$ mixing ratios
and trends, as compared to $CEDS_{2016}$. However, recent study reported a
difference in aviation emission distribution of $NO_x$ between CMIP5 and CMIP6
related to an error in data pre-processing in CEDS, leading to a northward shift
of $O_3$ burden in CMIP6 (Thor et al., 2023). Therefore, the contribution of the
aircraft emissions of $NO_x$ to the $O_3$ mixing ratios could be overestimated at high
latitude regions.
Surface $O_3$ measurements in the U.S. are obtained from the U.S.
Environmental Protection Agency (EPA). Linear trends of surface $O_3$ are
calculated separately for boreal summer (June-July-August, JJA) and winter
(December-January-February, DJF). Seasonal mean for any site that has less
than 50% data availability in any month of a season is discarded following Lin
et al. (2017). $O_3$ trends is calculated only when the seasonal data availability is
greater than 85% during the analyzed period (more than 22 years). Trends in
this study are calculated based on the linear least-squares regressions and the
statistical significance is identified through the F test with the 95% confidence
level.
**2.4 Experimental Design**
In this study, four groups of experiments are conducted, each group
includes both $NO_x$ tagging simulation and reactive carbon tagging simulation.
Two BASE experiment groups include simulations with emission sectors and
regions, respectively, tagged for the two chemical distinct precursors. The
BASE experiments are performed with time-varying anthropogenic emissions
and winds nudged to MERRA-2 reanalysis. The other two groups of sensitivity
experiments (MET) are the same as BASE experiments, except that the
anthropogenic emissions are held at year 2019 level during simulations. All
experiments are performed over 1990–2019, with the first 5 years treated as
model spin-up and the last 25 years used for analysis. The BASE experiments
are analyzed to quantify the source attributions of $O_3$ in the U.S., unless stated
otherwise. We note that although the wind fields are nudged at a 6-hourly
relaxation timescale, the atmospheric dynamics could also be slightly different
between simulations, leading to the slight changes in the contributions from the
same tags between simulations.
**2.5 Model Evaluation**
Figure 2 compares the simulated near-surface $O_3$ mixing ratios with those
from observations in 1995 and 2019, respectively. In general, the model
overestimates $O_3$ mixing ratios in the U.S. in both summer and winter by 10–
40%. It can capture the seasonal pattern of $O_3$ that high mixing ratios in summer
and low mixing ratios in winter. The spatial distributions can also be roughly
captured by the model, with statistically significant correlation coefficients
between simulations and observations in the range of 0.21–0.45. From 1995 to
2019, the $O_3$ mixing ratios in the U.S. decreased in summer and increased in
winter presented in observations. The model can produce the sign of the
changes, but has large biases in magnitudes, which will be discussed in the
following section.

**3 Results**
**3.1 Ground-level ozone trends in the U.S.**
Emissions of $O_3$ precursors have substantially reduced since 1995 in both
the western U.S. (WUS, 100–125°W, 30–45°N) and eastern U.S. (EUS, 70–
100°W, 30–45°N), primarily owning to the reductions in anthropogenic
emissions (Figs. S1–S3). However, the simulated annual near-surface $O_3$
mixing ratios present opposite trends between WUS and EUS, with increases
in EUS but weak decreases in WUS, which also exist in observations (Fig. 3a).

The simulated contrasting trends in annual mean $O_3$ mixing ratios between

the WUS and EUS are dominated by the strong decreases in $O_3$ mixing ratios
in summer across the U.S. (Fig. 3b) and increased $O_3$ levels in winter over the
central-eastern U.S. during 1995–2019 (Fig. 3c). The opposite trends between
summer and winter have also been noted in many previous studies (e.g.,
Copper et al., 2012; Lin et al., 2017, Jaffe et al., 2018). The model reproduces
the observed $O_3$ trend over EUS in summer and roughly captures the $O_3$ trend
over WUS in winter (Table 1). The decreasing trend over WUS in summer and
increasing trend over EUS in winter, however, are largely overestimated in the
model, partly attributed to the coarse model resolution. The model also tends
to overestimate the weakening of $NO_x$ titration in winter, leading to the biases.
For spring and autumn, they are the transition between summer and winter,
having the similar spatial pattern of $O_3$ trends as annual average, and will not
be concerned in this study.
**3.2 Source attribution of ozone trends to emission sectors**

During 1995–2019, summer and winter $NO_x$ emissions from energy and

surface transport sectors have significantly decreased in both WUS and EUS,
followed by industry and residential sectors, while those from aircraft have
increased slightly (Fig. 4). Emissions of NMVOCs from surface transportation,
solvents, industry, residential and waste sectors have decreased across the
U.S., while those from energy and agriculture have increased. CO emissions
have also significantly decreased over this time period.

The time series of the source sector contributions to $O_3$ mixing ratios from

$NO_x$ and reactive carbon emissions are shown in Fig. 5 and the $O_3$ trends in the
U.S. attributed to different emission source sectors are shown in Fig. 6. In
summer, the $O_3$ attributed to $NO_x$ emissions from energy and surface
transportation decreased at the rate of 2.0±0.2 and 1.6±0.2 ppb/decade in WUS
and 3.2±0.2 and 1.7±0.2 ppb/decade in EUS, respectively (Figs. 6a and 6c).
On the contrary, the $O_3$ contributed by aircraft $NO_x$ emissions increased by 0.4
± 0.0 ppb/decade in both WUS and EUS. Along with the reductions in
anthropogenic emissions, natural emissions are becoming increasingly
important as sources for $O_3$ formation near the surface. Although $NO_x$
emissions from soil are held at the present-day climatological levels, they
account for 0.7±0.1 and 1.7±0.1 ppb/decade increase in WUS and EUS,
respectively, during 1995–2019, related to the changing $O_3$ production
efficiency under the more $NO_x$-sensitive condition. Note that, during 1995–2019,
the molar ratio (mol N /mol C) of emitted $NO_x$ to NMVOCs reduced from 0.11 to
0.07 in WUS and from 0.14 to 0.07 in EUS, confirming the enhanced $NO_x$-
sensitive condition during the analyzed time period. In recent decades, global
emissions from international shipping have increased rapidly (Eyring et al.,
2005; Müller-Casseres et al., 2021), but have declined near the coast of the
United States. Due to a strong chemical sink associated with photolysis of $O_3$
with subsequent production of hydroxyl radical (OH) from water vapor in
summer (Johnson et al., 1999), the effect of increased international shipping
emissions over the remote ocean regions on the continental U.S. was blunted.
But the increase in shipping emissions inland tends to increase $O_3$ mixing ratios
in eastern U.S. (Fig. S4).
In summer, biogenic sources dominate the emissions of NMVOCs in the
U.S. (Fig. S3). As the $O_3$ decreases, mainly due to the reductions in domestic
$NO_x$ emissions, the contributions from biogenic emissions of NMVOCs have a
decreasing trend in the U.S. during 1995–2019 (Figs. 6b and 6d), even though
biogenic emissions were fixed during simulations. This also applies to $CH_4$, of
which the mixing ratio was kept constant. This does not actually mean that $CH_4$
and biogenic NMVOCs themselves contributed to the overall $O_3$ trend through
changing the precursor levels since they were kept constant during simulations;
rather, mainly due to the reductions in $NO_x$ emissions, $O_3$ production efficiency
by reactive carbon species decreases, leading to decreasing trends of $O_3$
contribution by $CH_4$ and biogenic NMVOCs. In conjunction with $NO_x$ emission
reductions, decreases in NMVOCs emissions from surface transportation and
industry sectors contribute negative $O_3$ trends of $-0.3\pm0.0$ and $-0.1\pm0.0$
ppb/decade, respectively, in both WUS and EUS in summer (Figs 6b and 6d),
which are offset by the increases in NMVOCs emissions from energy and
agriculture sectors. Although the $O_3$ production efficiency of CO is relatively low,
the contributions of CO to $O_3$ mixing ratios largely decreased with trends of $-$
$0.6\pm0.1$ and $-0.5\pm0.1$ ppb/decade in WUS and EUS, respectively, due to the
massive reduction in anthropogenic emissions of CO (Fig. S1).
In winter, through the weakened $NO_x$ titration process (Gao et al., 2013;
Simon et al., 2015), the $NO_x$ emission control causes an increase in $O_3$ levels
during 1995–2019, especially the contribution from surface transportation
($0.4\pm0.0$ ppb/decade in WUS and $0.8\pm0.1$ ppb/decade in EUS) (Figs. 6e and
6g). Although aircraft $NO_x$ emissions slightly increased, $O_3$ attributed to aircraft
$NO_x$ emissions shows positive trends as large as $0.4\pm0.0$ and $0.6\pm0.0$
ppb/decade in WUS and EUS, respectively. It is because aircraft emissions are
injected directly into the upper troposphere and lower stratosphere in a low
ambient $NO_x$ condition and have a much higher $O_3$ enhancement efficiency
than surface emissions (Hodnebrog et al., 2011). It can be confirmed that the
$NO_x$ from aircraft contributes to the increase in $O_3$ mixing ratios at 250 hPa in
high latitude regions of the Northern Hemisphere during 1995–2019 (Fig. S5).
The decrease in near-shore shipping emissions weakened the $NO_x$ titration,
together with the weakened $O_3$ chemical sink from water vapor in winter,
leading to large increasing trends of $O_3$ by $0.8\pm0.1$ and $1.0\pm0.1$ ppb/decade,
respectively, in the WUS and EUS during 1995–2019. Although most natural
emissions do not change during the simulations, the net $O_3$ chemical production
is more sensitive to $NO_x$ under the emission control condition, resulting in the
increasing $O_3$ trends contributed by the soil and lightning $NO_x$ emissions. Due
to the weakened $NO_x$ titration in winter, the contribution of stratospheric
intrusion increases at a rate of 0.6±0.1 and 1.0±0.1 ppb/decade over WUS and
EUS, respectively, when stratospheric contribution to the near-surface $O_3$ is
relatively high (Butler et al., 2018). Along with the weakened $NO_x$ titration,
contributions from reactive carbon emissions to the near-surface $O_3$ in the U.S.
also increase for most species and sectors (Figs. 6f and 6h).
**3.3 Source attribution of ozone trends to emission regions**
Time series of the source region contributions to near-surface $O_3$ mixing
ratios are shown in Fig. 7 and the $O_3$ trends in the U.S. attributed to different
emission source regions are presented in Fig. 8. In summer, domestic
anthropogenic $NO_x$ emissions (excluding those from soil) within North America
account for 49% of the near-surface $O_3$ mixing ratio averaged over the U.S.
(WUS+EUS) in 1995–2019. The domestic emission reduction is the dominant
factor causing the decline in surface $O_3$ mixing ratios, with contributions of −
4.4±0.2 and −5.7±0.3 ppb/decade to the trends over WUS and EUS,
respectively, during 1995–2019 (Figs. 8a and 8c). Reductions in the NMVOCs
emissions from North American anthropogenic sources also decrease $O_3$
mixing ratios (Figs. 8b and 8d), accompanying with the domestic $NO_x$ emission
control. The increase in $NO_x$ emissions from Asia contributes 0.7±0.1
ppb/decade to the total $O_3$ increasing trend in WUS, partly offsetting the
negative impact of domestic emission reductions, but has a weak impact in EUS,
which is consistent with previous studies (Lin et al., 2017).
In winter, domestic anthropogenic $NO_x$ emissions only account for 19% of
the surface $O_3$ mixing ratio in the U.S. over 1995–2019, while $NO_x$ sources from
lightning, rest of the world (mainly from the international shipping), and Asia
contribute 17%, 14%, and 11%, respectively. $O_3$ from stratospheric intrusion
contributes 21% of the near-surface $O_3$ in the U.S. in winter. During 1995–2019,
the significant increase in wintertime surface $O_3$ mixing ratios are not directly
linked to the reductions in domestic anthropogenic emissions (Figs. 8e and 8g).
However, the domestic emission control weakens the $NO_x$ titration, resulting in
considerable increases in $O_3$ originating from the natural sources, including $O_3$
from stratospheric intrusion, lightning and soil emissions. The natural sources
combined contribute to positive $O_3$ trends of 1.2±0.2 and 2.4±0.3 ppb/decade
in WUS and EUS, respectively. If the $O_3$ increase is attributed to NMVOCs
emissions, the combined natural source contribution is even larger (1.4±0.2 in
WUS and 2.5±0.2 ppb/decade in EUS) (Figs. 8f and 8h). $O_3$ produced by $CH_4$
increases at rates of 1.3±0.1 and 2.1±0.1 ppb/decade in WUS and EUS,
respectively, due to the weakened $NO_x$ titration. Increases in aviation and
shipping emissions together explain the 1.2±0.1 and 1.5±0.1 ppb/decade of $O_3$
trends in WUS and EUS, respectively (Figs. 6e and 6g). Long-range transport
of $O_3$ produced from Asian $NO_x$ emissions enhances the wintertime $O_3$
increasing trends by 0.9±0.1 and 1.2±0.1 ppb/decade in WUS and EUS,
respectively, which are equally contributed by sources from East Asia, South
Asia, and Southeast Asia (Figs. 8e and 8g).
**3.4. Impact of variations in large-scale circulations on ozone trends**
Many studies have reported that $O_3$ spatial distribution is strongly
modulated by changes in large-scale circulations (e.g., Shen and Mickley, 2017;
Yang et al., 2014, 2022). Based on our MET experiments with anthropogenic
emissions kept unchanged, the changes in large-scale circulations show a
weak influence on the U.S. $O_3$ trends in summer (Fig. 9a) but cause a significant
$O_3$ rise in the central U.S. in winter (Fig. 9b). Averaged over the U.S., the near-
surface $O_3$ mixing ratio in winter increases at the rate of 0.7±0.3 ppb/decade
during 1995–2019 in MET experiments. It suggests that the variation in large-
scale circulations is responsible for 15% of the increasing trend in wintertime
$O_3$ mixing ratio by 4.7±0.3 ppb/decade in the U.S. during 1995–2019 simulated
in BASE experiment.
The changes in atmospheric circulation pattern support the above finding.
Compared to 1995–1999, anomalous northerly winds locate over high latitudes
of North America in 2015–2019 (Fig. 9c), strengthening the prevailing northerly
winds in winter. In addition, an anomalous subsidence occurs over the central
U.S. in 2015–2019, compared to 1995–1999 (Fig. 9d). The anomalous
subsidence transport $O_3$ from high altitudes and even stratosphere to the
surface and the strengthened winds transport $O_3$ from remote regions (e.g., $O_3$
produced by Asian $NO_x$ emission) to the central U.S., both contributing to
0.2±0.1 ppb/decade of the $O_3$ increase over the U.S. (Fig. 10). The finding is
consistent with Lin et al. (2015) that variations in the circulation facilitate $O_3$
transport from upper altitudes to the surface, as well as foreign contributions
from Asia. The anomalous atmospheric circulation is likely linked to the location
of the midlatitude jet stream, which is influenced by ENSO cycle.

**4. Conclusions and discussions**
Using a global chemistry–climate model equipped with an $O_3$ source
tagging technique, we examine the long-term trends and source apportionment
of $O_3$ in the continental U.S. over 1995–2019 to various emission source
sectors and regions in this study. This model can capture the $O_3$ decreasing
trend over the EUS in summer and increasing trend over the WUS in winter
during this time period, but largely overestimates the decreasing trend over
WUS in summer and increasing trend over EUS in winter.
In summer, our simulation results show that the decline in surface $O_3$ is
dominated by the rapid reductions in $NO_x$ emissions from energy and surface
transportation sectors, contributing to $O_3$ decreases at a rate of –2.0 and –1.6
ppb/decade in WUS and –3.2 and –1.7 ppb/decade in EUS, respectively. As
the anthropogenic $NO_x$ decreases, the more $NO_x$-sensitive condition leads to a
positive $O_3$ trend of 0.7 and 1.7 ppb/decade in WUS and EUS, respectively,
contributed by the $NO_x$ emissions from soil. Due to the reductions in $NO_x$
emissions, the $O_3$ production efficiency by reactive carbon species also
decreased, leading to the decreasing contributions to $O_3$ from reactive carbon
species in summer during 1995–2019. Even though biogenic NMVOCs
emissions and $CH_4$ mixing ratio were fixed during simulations, their
contributions also decreased related to the weakened $O_3$ production efficiency
by these precursors. Source region tagging suggests that the domestic
emission reductions are primarily responsible for the decreasing trend in
summertime near-surface $O_3$ mixing ratios in the U.S. during 1995–2019.
The mechanisms of wintertime $O_3$ increases over the U.S. are more
complicated. First, the domestic emission control weakens the $NO_x$ titration,
resulting in considerable increases in $O_3$ originating from natural sources,
including $O_3$ from stratospheric intrusion, lightning, soil and biogenic emissions.
The natural sources combined contribute a positive $O_3$ trend of more than 1 and
2 ppb/decade in WUS and EUS, respectively. Second, increases in aviation and
shipping emissions together explain the 1.2 and 1.5 ppb/decade of $O_3$ trends in
WUS and EUS, respectively. Third, long-range transport of $O_3$ produced from
Asian $NO_x$ emissions enhances the wintertime $O_3$ increasing trends by 0.9 and
1.2 ppb/decade in WUS and EUS, respectively. Fourth, the variation of
horizontal and vertical transport $O_3$ associated with the changes in large-scale
circulation contributes to the near-surface $O_3$ increases over the U.S. by 15%
in winter during 1995–2019.
Compared to observations, the decreasing trend of $O_3$ mixing ratios over
WUS in summer and increasing trend over EUS in winter are overestimated in
the CAM4-chem model. Because most $O_3$ monitors are located in urban areas
and these areas generate strong $O_3$ during the day and have strong oxidation
titration at night, the daily and grid averaged $O_3$ mixing ratios output by the
model could be inconsistent with the urban observations. The overestimate of
$O_3$ trend in the EUS might be related to a potential biased model representation
of vertical mixing in winter. Large uncertainties existing in the emissions also
result in the biases in the $O_3$ simulation. Lin et al. (2017) found that the
contribution from increasing Asian emissions offset that from the U.S. emission
reductions, resulting in a weak $O_3$ trend in WUS. In this study, the Asian $NO_x$
emissions only contribute to 0.6 ppb/decade of the total positive trend in WUS
in summer, much lower than the 3.7 ppb/decade decrease attributable to the
domestic emission reductions, suggesting that the Asian contribution to the $O_3$
trends in WUS is possibly underestimated in this study. We also found that the
model did not capture the significant increase in summertime $O_3$ levels in China
in recent years, which could explain the low contribution from Asian sources.
Additionally, international shipping can have a disproportionately high influence
on tropospheric $O_3$ due to the dispersed nature of $NO_x$ emissions (Butler et al.,
2020; Kasibhatla et al., 2000; von Glasow et al., 2003), together with the
weakened $NO_x$ titration, resulting in the overestimation of $O_3$ trends. The fixed
$CH_4$ mixing ratio during simulations also biased the modeled $O_3$ trends, which
deserves further investigation with the varying $CH_4$ levels in future studies. The
coarse model resolution also contributed to the biases. The overestimate of $O_3$
trend over EUS in winter, likely related to the bias in $NO_x$ titration, implies the
overestimate of source contributions to the trends in magnitude.

Compared with Butler et al. (2018), the simulation in this study shares

similar source sector contributions to the zonal average of $O_3$ mixing ratios at
the surface and 400 hPa in 2010 (Figs. S7 and S8 in this study and Figs. 5 and
6 in Butler et al. (2018)). The contributions from the stratosphere and lightning
$NO_x$ are relatively higher in this study than Butler et al. (2018). This may be
related to the different anthropogenic emission inventories used, causing
different $O_3$ production/loss efficiencies by natural precursors. When comparing
the contributions from different source regions to surface $O_3$ mixing ratios in
North America, $NO_x$ emissions from East Asia, South Asia, North America, and
Europe contributed 2.2, 1.1, 8.3, and 0.7 ppb of the surface $O_3$ in North America,
respectively (Fig. S9) in this study, which are also similar to those from Fig. 4 in
Butler et al. (2020). Both studies show the contributions of anthropogenic
NMVOCs to surface $O_3$ mixing ratios in North America are less than 10 ppb.


***Author contributions.*** YY designed the research; PL and SL performed
simulations; PL analyzed the data. All authors including HW, KL, PW, BL, and
HL discussed the results and wrote the paper.

***Code and data availability.*** The CESM is maintained by NCAR and is provided
freely to the community. The ozone tagging code has been described by Butler
et al. (2018). The MERRA-2 reanalysis data are from NASA GESDISC data
(https://goldsmr5.gesdisc.eosdis.nasa.gov/data/MERRA2/M2I6NVANA.5.12.4/,
last access: 1 August 2022). The surface $O_3$ measurements in U.S. are
obtained from the U.S. Environmental Protection Agency
(https://aqs.epa.gov/aqsweb/airdata/download_files.html#Daily, last access: 1
August 2022). The modeling results are made available at
https://doi.org/10.5281/zenodo.6891316 (last access: 1 August 2022).

**Acknowledgments**
HW acknowledges the support by the U.S. Department of Energy (DOE), Office
of Science, Office of Biological and Environmental Research (BER), as part of
the Earth and Environmental System Modeling program. The Pacific Northwest
National Laboratory (PNNL) is operated for DOE by the Battelle Memorial
Institute under contract DE-AC05-76RLO1830.

***Financial support.*** This study was supported by the National Key Research
and Development Program of China (grant 2020YFA0607803), Jiangsu
Science Fund for Distinguished Young Scholars (grant BK20211541) and the
Jiangsu Science Fund for Carbon Neutrality (grant BK20220031).

***Competing interests.*** The authors declare that they have no conflict of interest.

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

**Table 1.** $O_3$ trends (ppb/decade) over eastern U.S. and western U.S. in winter
(December-January-February, DJF) and summer (June-July-August, JJA) from
observations and model simulations.

| Season | Source | eastern U.S. | western U.S. |
|--------|--------|--------------|--------------|
| DJF | Observation | 2.1 ± 0.29 | 2.2 ± 0.23 |
| DJF | Model | 6.1 ± 0.40 | 3.2 ± 0.28 |
| JJA | Observation | -3.0±0.41 | -0.5 ± 0.42 |
| JJA | Model | -3.0±0.29 | -2.3 ± 0.20 |


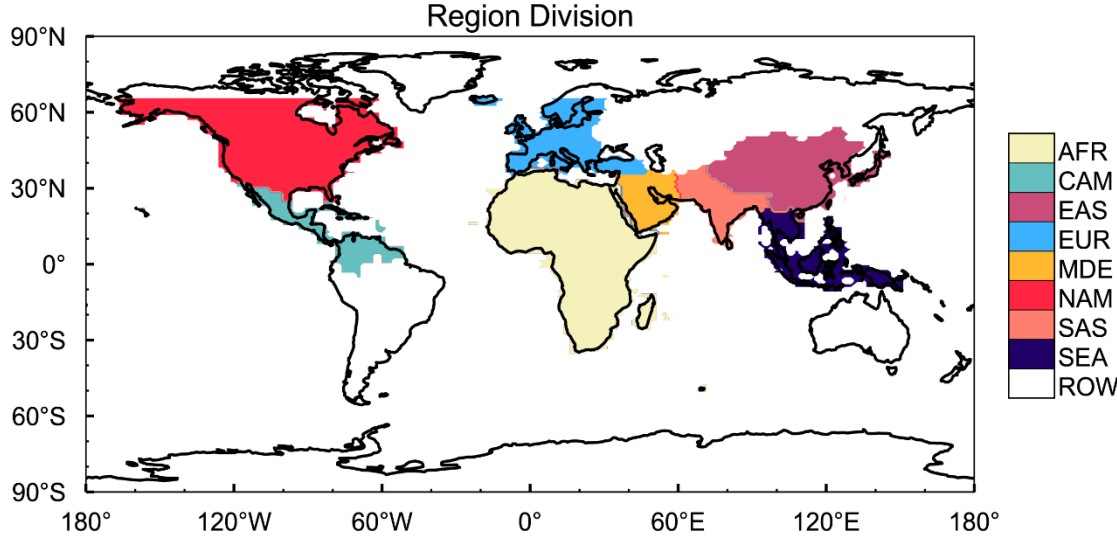

**Figure 1.** Source regions that are selected for $O_3$ source tagging in this study, include Africa (AFR), Central America (CAM), East Asia (EAS), Europe (EUR), Middle East (MDE), North America (NAM), South Asia (SAS), Southeast Asia (SEA) and rest of the world (ROW).

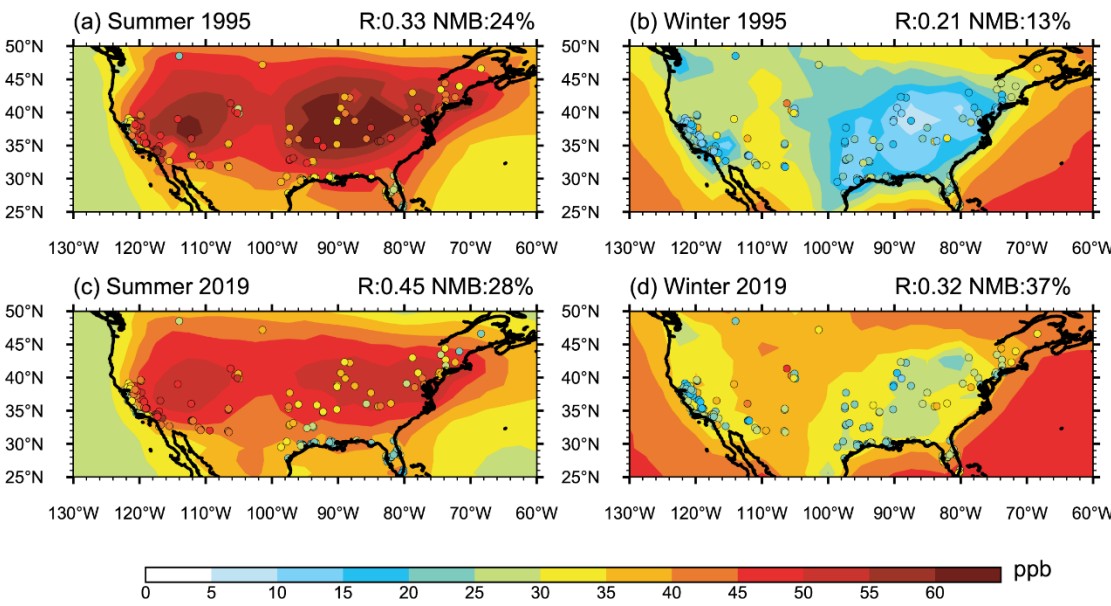

**Figure 2.** The simulated (contours) and observed (scatters) seasonal mean near-surface O3 mixing ratios over the United States in JJA (left) and DJF (right) and in 1995 (top) and 2019 (bottom). The correlation coefficient and normalized mean bias (NMB, $\sum$ (Model − Observation) / $\sum$ Observation $\times$ 100%) are shown on top right of each panel.

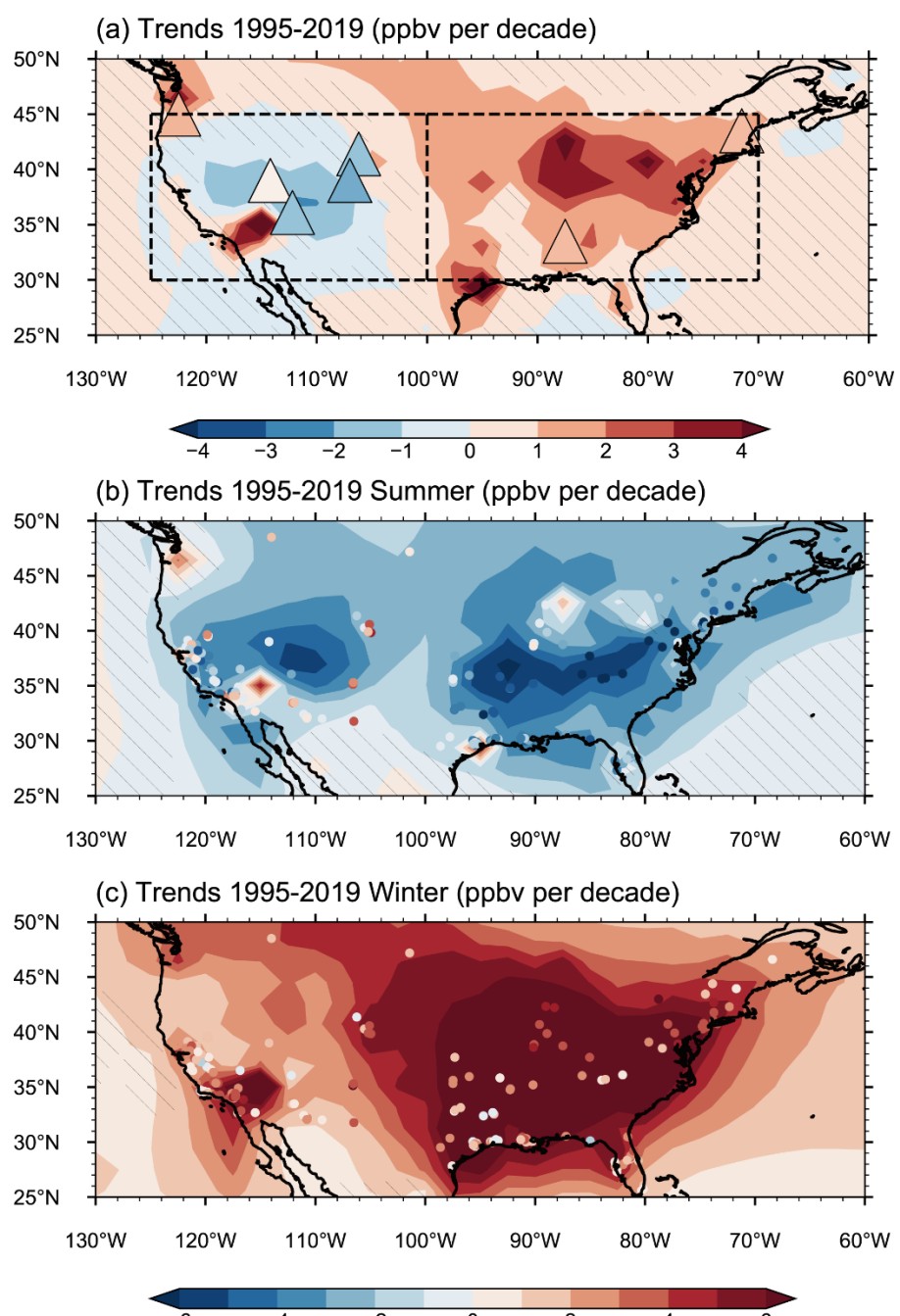

926
927
**Figure 3.** Linear trends (ppb/decade) of simulated (contours) and observed
(color-filled markers) (a) annual, (b) JJA and (c) DJF mean near-surface $O_3$
mixing ratios during 1995–2019. Areas without hatches indicate statistical
significance with 95% confidence. The boxes in (a) mark the western U.S.
(WUS, 100–125°W, 30–45°N) and eastern U.S. (EUS, 70–100°W, 30–45°N),
respectively. The observed annual $O_3$ mixing ratio trends in (a) are derived from
IPCC AR6, based on Cooper et al. (2020) and Gaudel et al. (2020) over 1995–
2017. The observed seasonal $O_3$ mixing ratio trends in (b) and (c) are calculated
based on the U.S. EPA $O_3$ measurements over 1995–2019.


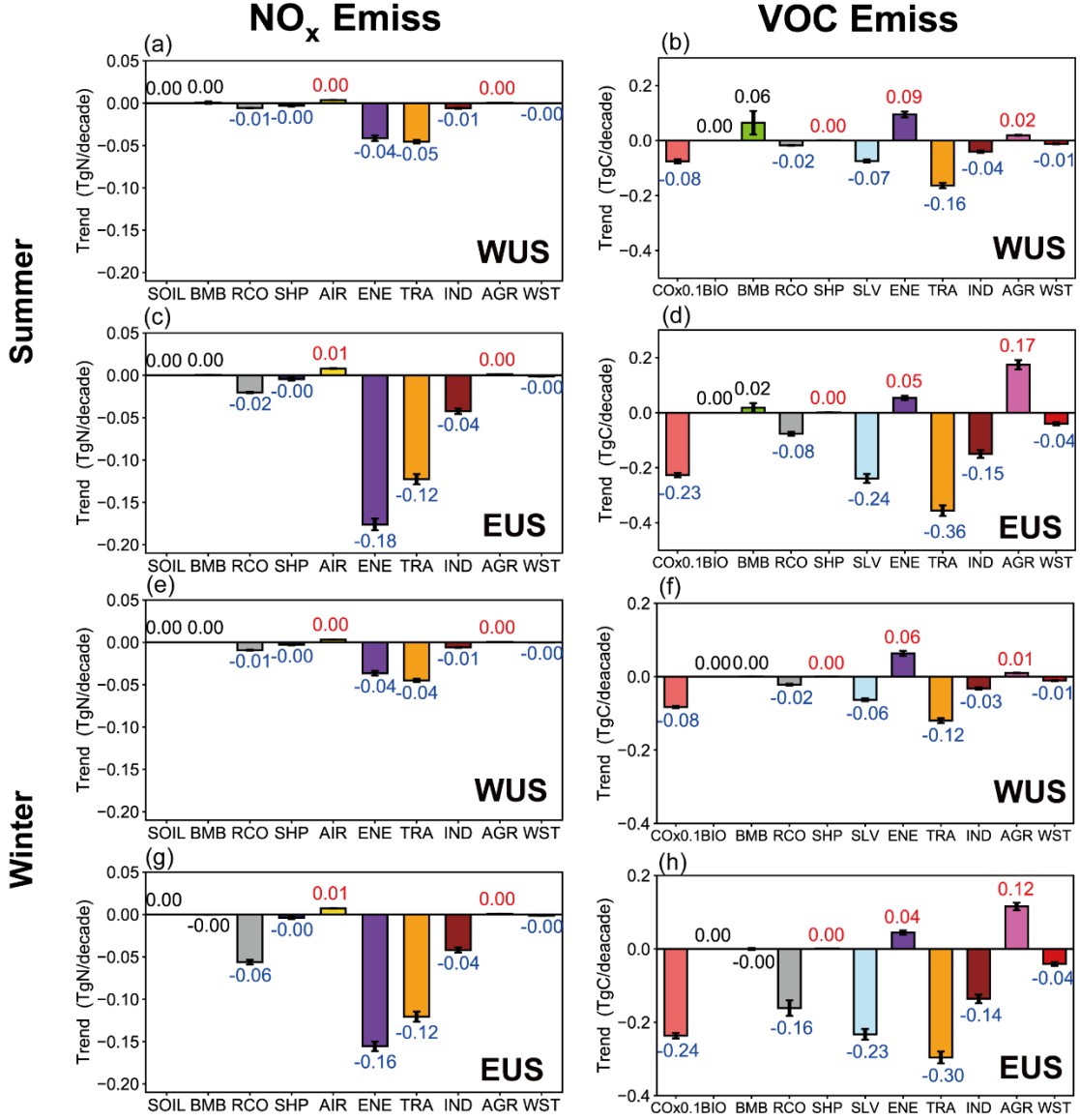



**Figure 4.** Linear trends of NOx and reactive carbon emissions from various
sectors in summer and winter over WUS and EUS. The increasing and
decreasing trends marked with red and blue values, respectively, indicate
statistical significance with 95% confidence.

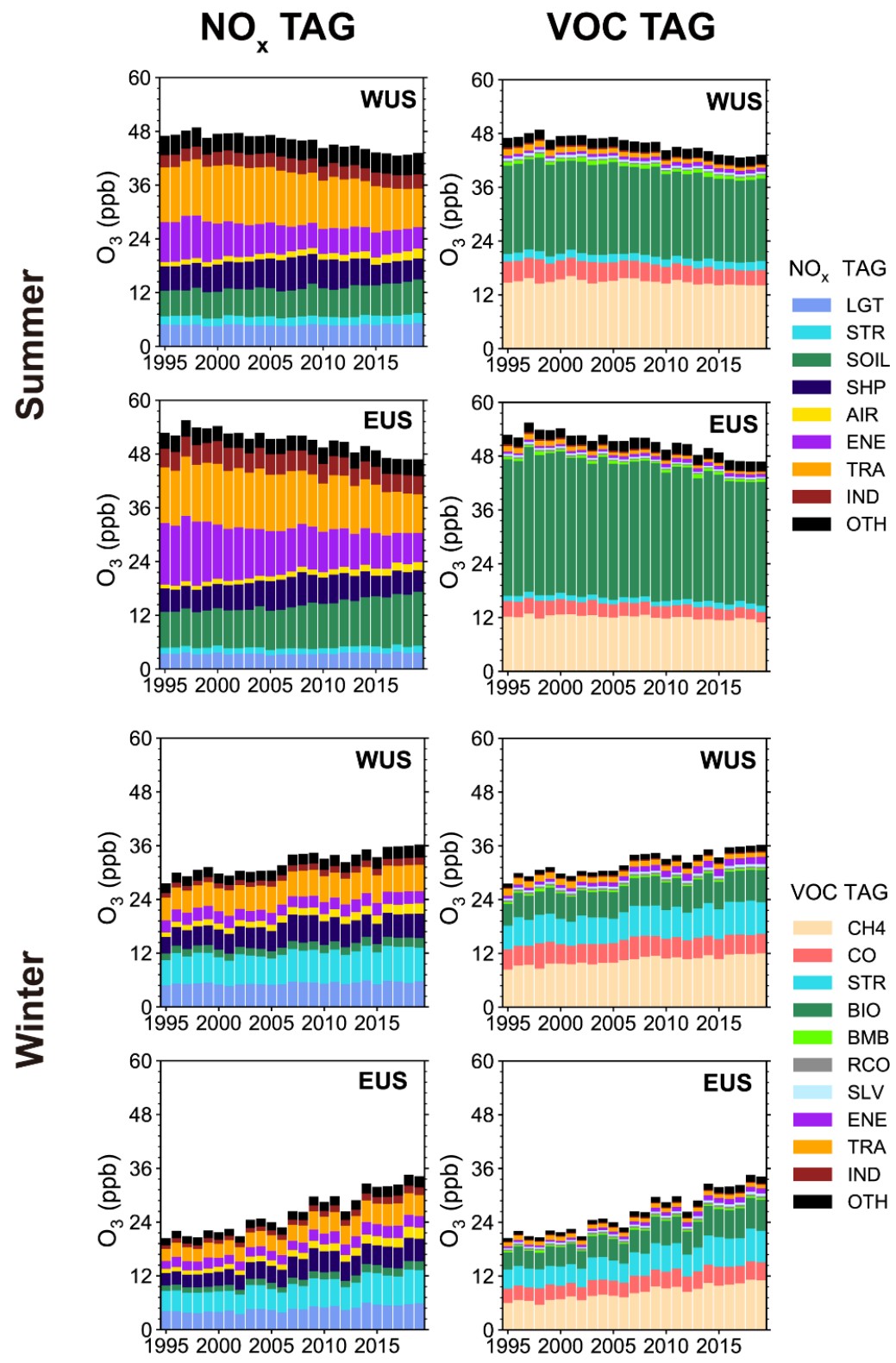



**Figure 5.** Time series of near-surface $O_3$ mixing ratios (ppb) averaged over
WUS and EUS contributed by $NO_x$ and reactive carbon emissions from
different sectors in summer and winter during 1995–2019. Sources with small
contributions are combined and shown as OTH.

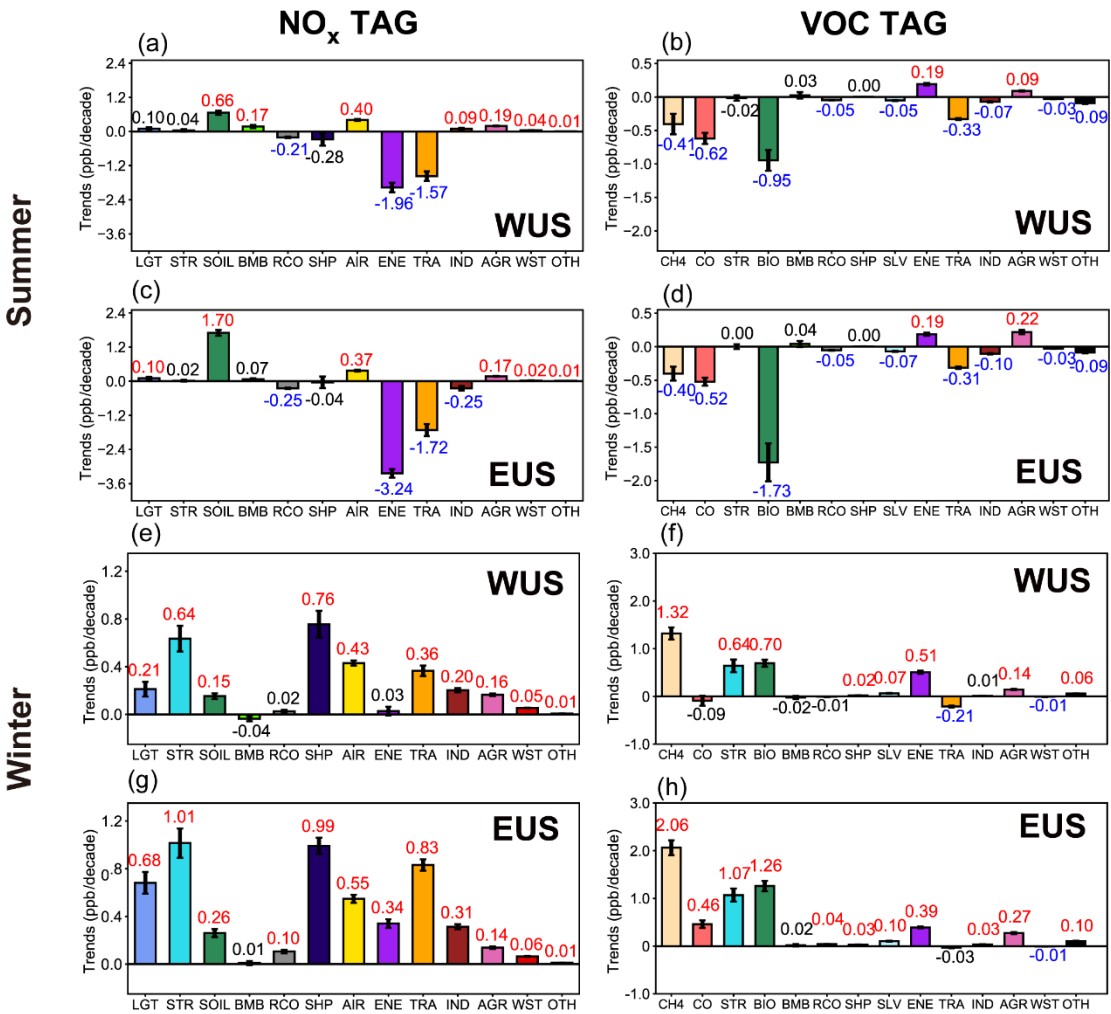

**Figure 6.** Linear trends (ppb/decade) of near-surface O₃ mixing ratios in summer and winter over WUS and EUS contributed by the NOₓ (left) and reactive carbon (right) emissions from various sectors (color bars). The increasing and decreasing trends marked with red and blue color numbers, respectively, indicate statistical significance with 95% confidence. Other sources having small contributions are combined and shown as OTH.

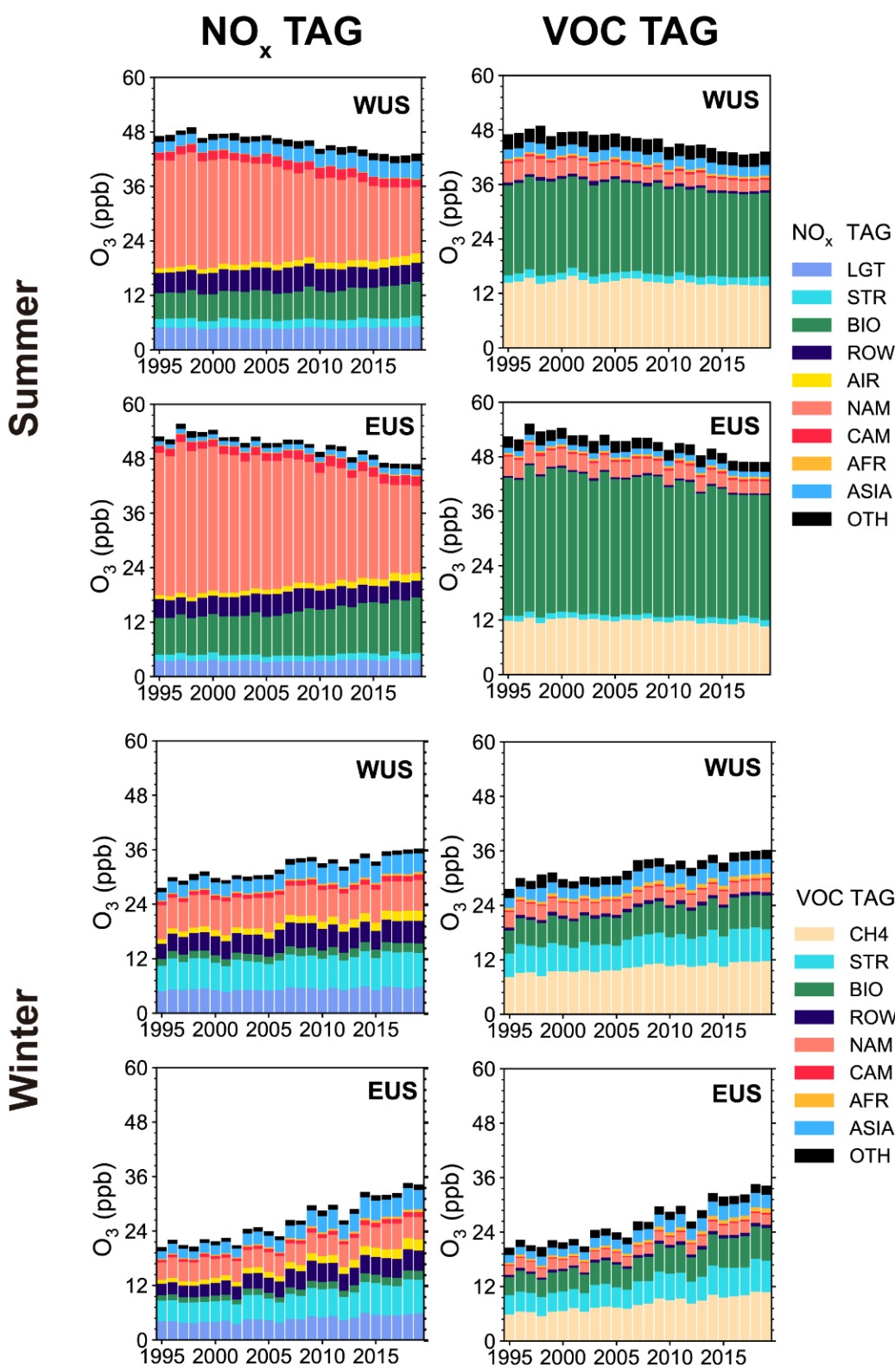

**Figure 7.** Time series of near-surface $O_3$ mixing ratios (ppb) averaged over
WUS and EUS contributed by $NO_x$ and reactive carbon emissions from different
source regions in summer and winter during 1995–2019. Sources with small
contributions are combined and shown as OTH.

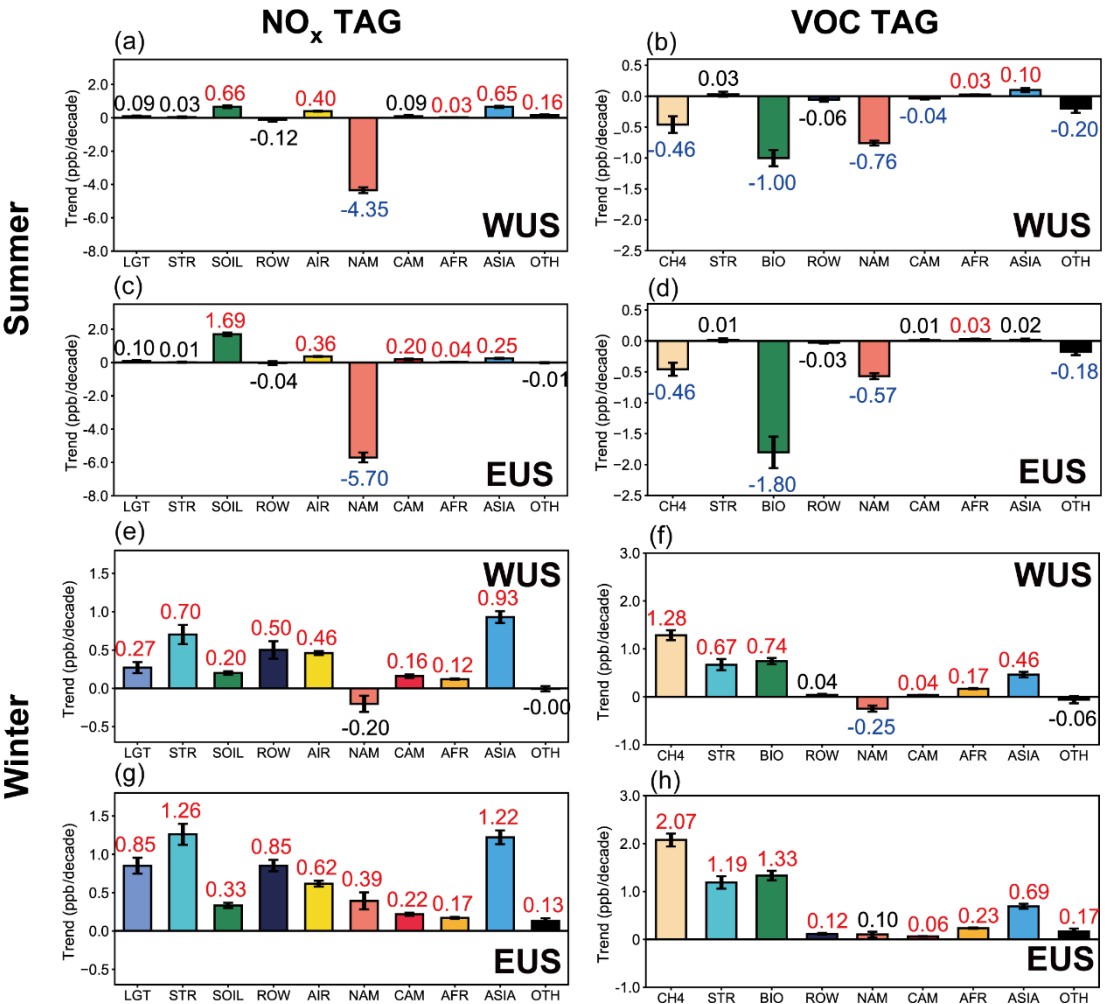

**Figure 8.** Linear trends (ppb/decade) of near-surface O₃ mixing ratios in summer and winter over WUS and EUS contributed by the NOₓ (left) and reactive carbon (right) emissions from various source regions (color bars). The increasing and decreasing trends marked with red and blue color numbers, respectively, indicate statistical significance with 95% confidence. Contributions from source regions EAS, SAS and SEA are combined to ASIA. Other sources having small contributions are combined and shown as OTH.

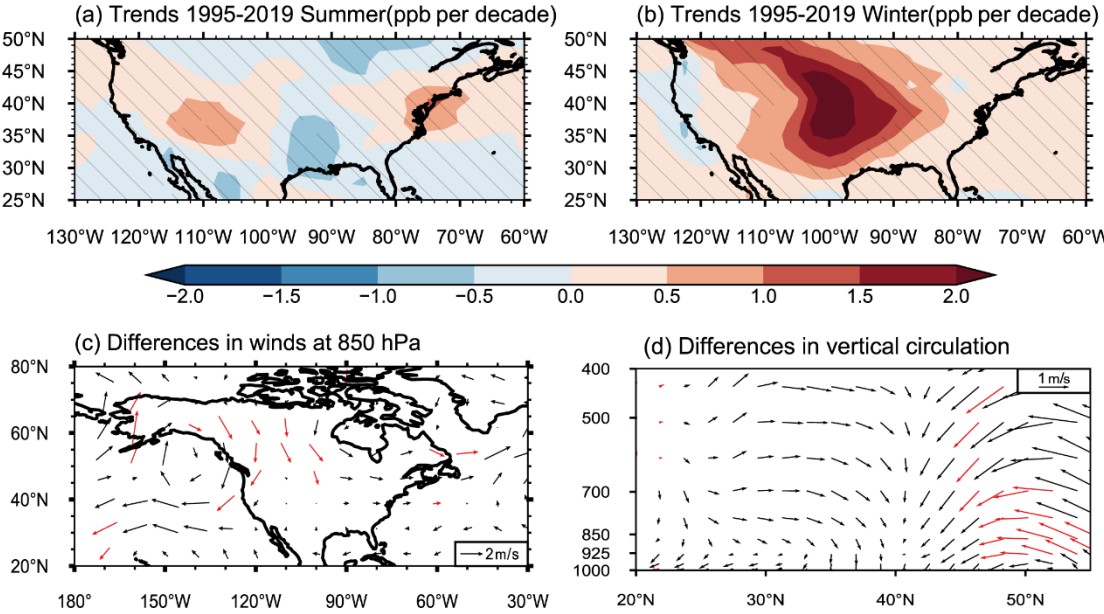

**Figure 9.** Linear trends (ppb/decade) of simulated (a) JJA and (b) DJF mean near-surface $O_3$ mixing ratios during 1995–2019. Differences between the first (1995–1999) and last (2015–2019) five years during 1995–2019 (last–first) in DJF mean (c) 850 hPa horizontal winds and (d) meridional winds and vertical velocity averaged over 90–105°W. Areas without hatches in (a) and (b) and red arrows in (c) and (d) indicate statistical significance with 95% confidence. All results are from the MET experiments.

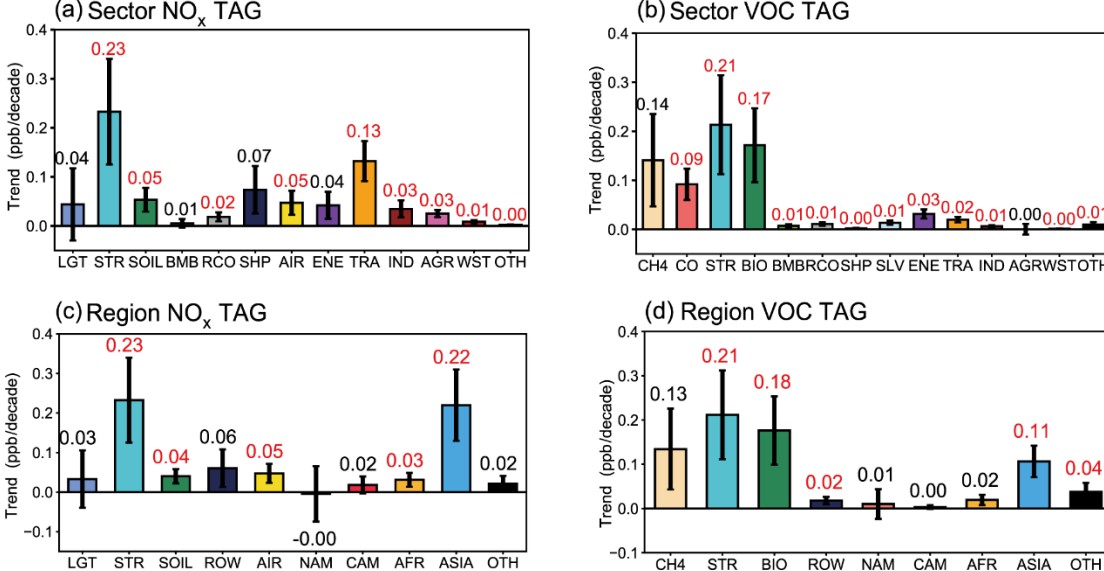

**Figure 10.** Linear trends (ppb/decade) of near-surface O₃ mixing ratios in winter over the U.S, contributed by the NOₓ (a,c) and reactive carbon (b,d) emissions from various source sectors (a,b) and regions (c,d). The increasing and decreasing trends marked with red and blue color numbers, respectively, indicate statistical significance with 95% confidence. Contributions from source regions EAS, SAS and SEA are combined to ASIA. Some sources having small contributions are combined and shown as OTH.