# Peer review of "Source attribution of near-surface ozone trends in the United States during 1995–2019"

_Atmospheric Chemistry and Physics, 2022_

## Author Comment (AC1)

**Manuscript # acp-2022-678**

**Responses to Referee #1**

This manuscript details a multi-year ozone tagged contribution analysis. The specific value is the attempt to explain observed trends with trends of model contributions. I think this manuscript has high value, but needs some additional analysis to be published. Below are sections that summarize comments related to model performance, methods, and editorial notes. Last, is a line-by-line section that has more specific feedback.

We thank the editor for all the insightful comments. Below, please see our point-by-point response (in blue) to the specific comments and suggestions and the changes that have been made to the manuscript, in effort to take into account all the comments raised here.

Generally, this manuscript is missing basic model evaluation in the body and supplement. The current manuscript jumps into trends of contributions and only mentions evaluation in the conclusions. In particular, only evaluation in China is ever discussed. The manuscript focuses on trends associated with titration without ever demonstrating the model reasonably captures the phenomenon. The representation of titration in the Eastern US by the model is important given that it drives the trends. Given that coarse models (2x2.5 degree) are often extremely biased at nighttime, the authors should provide some evidence that nighttime titration is reasonably simulated and/or describe how model artifacts may play a role in the trend. Overall, it seems odd that performance over China is used to suggest underestimation of long-range transport while the performance and possible errors of local contributions are omitted.

Response:
    We have now added a figure and corresponding descriptions for the model evaluation. The model also tends to overestimate the weakening of $NO_x$ titration in winter, leading to the biases in trends in winter. We also removed the performance for China. Please see our responses below.

In the methods sections, more detail is needed on several fronts. The emissions are currently under-described even though they are well referenced. Recommendations are made in the line-by-line section. Similarly, the model-observation pairing is mostly left to the reader to infer. Again, recommendations are made in the line-by-line.

Response:
    Thanks for the suggestion. Please see the line-by-line responses below.

Especially in the conclusions, there are several statements where increase/decrease seem to be used incorrectly. These incorrect directional statements should be corrected. See lineby-line section for specific recommendations.

The section describing the overall trends would benefit from a table. The descriptions and parenthetical references make it somewhat difficult to easily compare. See line-by-line section for specific recommendations.

Lastly, recent application of a very similar system was made by Butler et al. 2020. That publication is referenced, but more should be done to compare these methods and results to those. From a methods standpoint, it would be nice to provide a short summary that explains how these experiments are different. From a results standpoint, you should compare the overlapping 2010 year for comparable. Are the results comparable for overlapping (2010) or proximate years? If not, do methodological differences explain discrepancies?

Response:
    Thank you for your suggestion. We corrected statements and added tables to describe overall trends.
    We compare the 2010 results of our simulations (Figs. S7–S9) with Butler et al. (2018, 2020) and added the following in the discussion: "Compared with Butler et al. (2018), the simulation in this study shares similar source sector contributions to the zonal average of $O_3$ concentrations at the surface and 400 hPa in 2010 (Figs. S7 and S8 in this study and Figs. 5 and 6 in Butler et al. (2018)). The contributions from the stratosphere and lightning $NO_x$ are relatively higher in this study than Butler et al. (2018). This may be related to the different anthropogenic emission inventories used, causing different $O_3$ production/loss efficiencies by natural precursors. When comparing the contributions from different source regions to surface $O_3$ concentrations in North America, $NO_x$ emissions from East Asia, South Asia, North America, and Europe contributed 2.2, 1.1, 8.3, and 0.7 ppb of the surface $O_3$ in North America, respectively (Fig. S9) in this study, which are also similar to those from Fig. 4 in Butler et al. (2020). Both studies show the contributions of anthropogenic NMVOCs to surface $O_3$ concentrations in North America are less than 10 ppb."

* 117, please describe the depth of the first layer and the number of layers in near the surface (e.g., under 2km). This helps contextualize the model representation of titration later.

Response:
    Thanks for the suggestion. We have now included such context as follows: "The height of bottom layer near the surface is about 120 m and there are about 4 layers within2 km."

* 121, please describe how the stratospheric values are set. Are they based on climatological values? Are they scaled based on something?

Response:
     Yes, stratosphere-troposphere exchange of $O_3$ is treated by setting $O_3$ to stratospheric values as their climatological means over 1996–2005 at the tropopause (Lamarque et al., 2012), which is affected by atmospheric circulation and experiences the same loss rates as $O_3$ in the troposphere (Tilmes et al.,2016). We have revised our description in the manuscript.

* 146, are XTR tags really neither NOx nor VOC? Are they included in both?

Response:
     This is a special kind of tagging, and its use is usually due to the fact that we cannot attribute it well to the source of the currently running tagging system, none of the reactants belong to the $O_x$ chemical family resulting in no tags can be passed to the $O_3$. Some examples are as follows: When $NO_x$ is being tagged, the reactions of $HO_2$ with certain organic peroxy radicals produce $O3\_X\_XTR$. A reaction during VOC tagging is the production of the specially tagged species $HO_2\_X\_XTR$ from the reaction between OH and $H_2O_2$ (Butler et al., 2018). So XTR exists in both.

* 159, It says CO and CH4 are not tagged by individual sources? Does that mean just by regions? Or, all CO is lumped? The wording is currently unclear. Particularly interested for CO.

Response:
     We have clarified as "We does not tag $CH_4$ by individual sources and its contribution is lumped, because $CH_4$ is often considered separately from NMVOCs. It has a relative long lifetime in the troposphere and it is well mixed in the troposphere due to its exceptionally low reactivity, which can contribute to $O_3$ formation at any location in the troposphere where photochemical conditions are favorable (Fiore et al., 2008). CO also has a longer lifetime and lower reactivity than most NMVOCs, separately tagging of CO is more conducive to distinguish its contribution to $O_3$ from other NMVOCs. Therefore, the lumped total CO is separately tagged in the sector attribution simulations, but the CO is not specifically tagged in the regional attribution simulations due to the computational limit."

* 164, as you note, this limitation of CO seems odd.

Response:
     We have removed this part. Please see our response above

* 168, It would be useful to note here (in addition to later) that TgN and TgC are shown in the appendix.

Response:
    Thank you for your suggestion. Added.

* 173, This seems like an important methodological shift. Can the authors highlight whether conclusions are robust to analysis from 1995-2015 or 1995-2019?

Response:
    The biomass burning only accounts for a very small amount of total $NO_x$ and NMVOCs emissions (Fig. S1). Therefore, the biomass burning emissions interpolated from SSP2-4.5 forcing scenario should not affect the results.

* 175, Can you clarify what "present-day" means here? Is this a climatology based on a range of "present-day" years or a specific year?

Response:
    Clarified as "… and are kept at the present-day (2000) climatological levels during simulations."

* 177, Please elaborate on Price parameterization. I think you are saying online parameterization based on simulated cloud top heights. There are also climatologies based on Price, so it is good to be clear.

Response:
    We added a further explanation of the parameterization: "Lightning emissions of $NO_x$ are estimated using online parameterization based on simulated cloud top heights from Price et al. (1997), which is scaled to provide a global annual emission of 3–5 Tg N $yr^{-1}$."

* 180-185, Please clarify whether the simulation is being sampled only at observation sitedays or averaged seasonally and then sampled at sites.

Response:
    The simulation is averaged seasonally and then sampled at the grid boxes of sites. For observations, seasonal mean for any site that has less than 50% data availability in any month of a season is not calculated. $O_3$ trends at sites is shown only when the data availability is greater than 85% during the analyzed period. Since the observational data are quality-controlled, we don't expect the seasonal average for simulations can largely influence the comparison, but the coarse model resolution may contribute to the biases when comparing with the

observations. We have now added this bias in the discussion.

* 200, The results should start with some estimate of model performance over the target analysis areas. At least, 1) a map of the model with obs scattered on it for an early year and a late year and 2) a description of how basic performance stats change over time. Because this paper focuses on the JJA and DJF, I would expect the model performance to have a similar separation. This will help the readers contextualize results.

Response:
 Thank you for the suggestion. We have now added the distribution of observed and modeled surface $O_3$ in the United States for summer and winter in 1995 and 2019 in Figure S4. We have also included a model evaluation section as:
 "Figure S4 compares the simulated near-surface $O_3$ concentrations with those from observations in 1995 and 2019, respectively. In general, the model overestimates $O_3$ concentrations in the U.S. in both summer and winter by 10–40%. It can capture the $O_3$ seasonality that high concentrations in summer and low concentrations in winter. The spatial distributions can also be roughly captured by the model, with statistically significant correlation coefficients between simulations and observations in the range of 0.21–0.45. From 1995 to 2019, the $O_3$ concentrations in the U.S. decreased in summer and increased in winter presented in observations. The model can produce the sign of the changes, but has large biases in magnitudes, which will be discussed in the following section."

[Figure]

**Figure S4.** The simulated (contours) and observed (scatters) seasonal mean near-surface $O_3$ concentrations over the United States in JJA (left) and DJF

(right) and in 1995 (top) and 2019 (bottom). The correlation coefficient and normalized mean bias (NMB, $\sum$ (Model − Observation) / $\sum$ Observation× 100%) are shown on top right of each panel.

* 205, Please add lightning NOx in the supplemental figures.

Response:
    The model did not output the lightning emission, but the emissions of NO from lightning are scaled to provide a global annual emission of 3–5 Tg N yr$^{-1}$ as Lamarque et. al. (2012).

* 214, related to 180-185, are these trends based on the model only at observation sites or based on averages of the regional "box"

Response:
    They are based on the regional grid boxes. The coarse model resolution may contribute to the model biases when comparing with the observations. We have now added this bias in the discussion.

* 214: Looking at Figure 4b, there is a lot of heterogeneity in the western summer trends. The Western cities are fairly isolated leading to misrepresentation by coarse global models. Can you discuss what would happen if you only looked at CASTNet or rural monitors? Or just the IPCC sites?

Response:
    I think you were referring to Figure 2b. For some heterogeneity in the summer trends, other studies also found $O_3$ increasing in Los Angeles and some cities in the central United States, and decreasing in Nevada and Utah based on observations at rural sites (Cooper et al., 2012; Lin et al., 2017). That may explain the strong decreasing trend over western U.S. produced by the model. We have revised the description as that "The decreasing trend over WUS in summer and increasing trend over EUS in winter, however, are largely overestimated in the model, partly attributed to the coarse model resolution."

* 216-223, I found it difficult to keep the text organized in my mind. I recommend adding a table here.
Response:
    We have added Table S1 to show the values.

**Table S1.** $O_3$ trends (ppb/decade) over eastern U.S. and western U.S. in DJF and JJA from observations and model simulations.

| Season | Source | eastern U.S. | western U.S. |
|--------|--------|--------------|--------------|
| DJF | Observation | 2.1 ± 0.29 | 2.2 ± 0.23 |
| DJF | Model | 6.1 ± 0.40 | 3.2 ± 0.28 |
| JJA | Observation | -3.0±0.41 | -0.5 ± 0.42 |
| JJA | Model | -3.0±0.29 | -2.3 ± 0.20 |

It would also be good to add some clarify on what "well produce" means Based on a 95% certainty, the CI are not overlapping for Eastern winter or Western summer. The CIs for Western winter are barely overlapping and only after rounding. The model seems to clearly reproduce the trend only for Eastern summer.

Response:

We have revised the description as "The model reproduces the observed $O_3$ trend over EUS in summer and roughly captures the $O_3$ trend over WUS in winter (Table S1). The decreasing trend over WUS in summer and increasing trend over EUS in winter, however, are largely overestimated in the model, partly attributed to the coarse model resolution."

For me, the titration performance in the East raises questions about the West. The model seems to dramatically overestimate the reduced titration in the East. Given the population density of the East, the titration is likely more widely spread. Due to the population sparsity of the West, the overestimated titration is likely diluted. How does this impact the conclusion about well representing winter in the West?

Response:

Thank you for pointing it out. We agree with the reviewer that the overestimation of $O_3$ trend in western U.S. in winter could be attributed to the overestimation of the weakened $NO_x$ titration. We have now added a note that "The model also tends to overestimate the weakening of $NO_x$ titration in winter, leading to the biases."

* 236, I am surprised to see STR (stratosphere) in both NOx and VOC. Is that via XTR?

Response:

The STR tag is neither from $NO_x$ nor VOCs emissions. In both $NO_x$ and VOCs tagging, initial conditions for $O_x$ species in the stratosphere were tagged with STR. In addition, the photolysis of $O_2$ and $N_2O$ ultimately produces the $O_3$,

which is all tagged as STR. They can be transported downward via atmospheric circulation and contribute to the near-surface $O_3$ concentrations.

* 241, can you add error bars to the figure?

Response:
    Yes, we have added it.

* 243-247, I think this is a very interesting finding! If the atmosphere is increasingly NOx sensitive, that should have important implications for VOC tagging in later years. Can you discuss that a bit more?

Response:
    Thanks for your suggestion, we have added the following: "Note that, during 1995–2019, the molar ratio (mol N /mol C) of emitted $NO_x$ to NMVOCs reduced from 0.11 to 0.07 in the WUS and from 0.14 to 0.07 in the EUS, confirming the enhanced $NO_x$-sensitive condition during the analyzed time period."

* 247-251, What role does the location of monitors play in the conclusion here? Is there a strong spatial gradient to the SHP contribution? This is important because the populations tend to be skewed toward near the ocean. In an ideal world, it would be interesting to see a few maps (1995 and 2019) of contributions trends that have a strong spatial gradient.

Response:
    Yes, the SHP contribution trends has a strong spatial gradient. We have added a figure below to describe the trend of shipping emissions and $O_3$ contribution, then modified the shipping-related part as follows:
    In recent decades, emissions from international shipping have increased rapidly (Eyring, 2005; Müller-Casseres et al., 2021), but have declined near the coast of the United States. Due to a strong chemical sink associated with photolysis of $O_3$ with subsequent production of hydroxyl radical (OH) from water vapor in summer (Johnson et al., 1999), the effect of increased emissions of the far-shore ocean on the continental United States was blunted. But the increase in shipping emissions inland tends to increase $O_3$ concentrations in eastern U.S.
    In winter, the decrease in near-shore shipping weakened the $NO_x$ titration, together with the weakened $O_3$ chemical sink from water vapor in winter, leading to large increasing trends of $O_3$ by 0.8±0.1 and 1.0±0.1 ppb/decade, respectively, in the WUS and EUS during 1995–2019.

[Figure]

**Figure S5.** Trends of shipping emissions of $NO_x$ and $O_3$ trends contributed by shipping emissions in JJA and DJF from 1995 to 2019.

* 259, I find this to be a particularly interesting finding that has implications for the estimation of climate/air quality co-benefit assessments. I wish it was expanded a bit in the conclusions.

Response:

Thank you for the suggestion. We have expanded it in the conclusions as follows:

"Due to the reductions in $NO_x$ emissions, the $O_3$ production efficiency by reactive carbon species also decreased, leading to the decreasing contributions to $O_3$ from reactive carbon species in summer during 1995–2019. Even though biogenic NMVOCs emissions and $CH_4$ concentrations were fixed during simulations, their contributions also decreased related to the weakened $O_3$ production efficiency by these precursors."

* 272, I find this confusing. Most of this sentence makes perfect sense to me. It is introduced, however, in the context of reduced VOCs. At aircraft heights, you say that only NOx increase. Does that mean that there are no aircraft VOCs? If so, are you suggesting that VOCs at 6-10km were meaningfully reduced and that contributed to the large aircraft trends?

Response:

We apology for the confusing. We were trying to say only the aircraft sector increased and other anthropogenic sectors decreased. By the way, the aircraft mainly emits $NO_x$ rather than NMVOCs. We have rephased the description as

that "Although aircraft $NO_x$ emissions slightly increased, but $O_3$ attributed to aircraft $NO_x$ emissions shows positive trends as large as 0.4±0.0 and 0.6±0.0 ppb/decade in WUS and EUS, respectively, because aircraft emissions are injected directly into the upper troposphere and lower stratosphere in a low ambient $NO_x$ condition and have a much higher $O_3$ enhancement efficiency than surface emissions (Hodnebrog et al., 2011)."

* 280-282, Similar comment to earlier. The spatial nature of this enhancement is important. It'd be great to see a map of the contribution and trends.

Response:
  Added and please see the response above.

* 283, I don't think this is strictly speaking true. Your lightning emissions are parameterized based on simulated clouds. Can you clarify that this is only true for VOC?

Response:
  $NO_x$ emission from the soil is fixed in our simulation, and lightning $NO_x$ varies but is not likely to have a clear trend. All natural VOCs emissions are hold constant during the simulations. We have revised the descriptions as "although most natural emissions do not change during the simulations, …".

* 289, Butler et al compared January and July in Figure 5. It isn't clear to me that the contribution maximized in DJF vs MAM.

Response:
  We have now revised the text as follows: "when stratospheric contribution to the near-surface $O_3$ is relatively high".

* 295, Specify anthropogenic and/or that you are excluding soil NOx. Soil NOx in summer has a large anthropogenic component and the contribution from soil is likely "domestic" (e.g. Lapina et al. 2014).

Response:
  We revised it to "domestic anthropogenic $NO_x$ emissions (excluding those from soil)".

* 316, This is definitely interesting... I'm struck however by the dramatic overestimate in the trend, which might be related to the models representation of vertical mixing in winter.

Response:
  Thank for your suggestion. We have added this possible explanation in the

conclusion as "The overestimate of O$_3$ trend in the EUS might be related to a potential biased model representation of vertical mixing in winter".

* 326-327, The idea that South Asia, and Southeast Asia East Asia "equally contribute" is a somewhat surprising finding. Many previous refereed articles show a decreased transport efficiency from India (S. Asia) to the US compared to East Asia. Similarly, Butler et al 2020 showed significantly larger East Asia contribution than South Asia. Can you highlight why your results would be so different?

Response:
Here the result shows their contribution to the O$_3$ trends not to O$_3$ concentrations. The reason for this phenomenon may be due to China's rapid reduction of NO$_x$ emissions in industry and energy in recent years. But we also noticed that the model can not well simulate the O$_3$ trends in China, which deserves further investigation in our future work.

* 335-350, This discussion really highlights the oversimplicity of linear trends. The authors do a good job noting this is likely associated with transport. It would be good if they connected it to known meteorological cycles. A quick look shows that ENSO cycles are likely accounted for by the time averaging, but the Pacific Decadal Oscillation (PDO) is not.
This highlights why the trend is likely not significant. It is likely made up of ups and downs seen in the PDO. A 5-year average of the NCEI PDO index shows that the winters of these two periods are of opposite signs (despite inter annual variability). This is in part because in mid 1998, the PDO index shifted. This leads to a smaller difference between summers than winters. You could also reference the Lin et al. paper about the position of the jet stream.

Response:
Thanks for your suggestion. We have now included such context as follows:" The horizontal and vertical transport of O$_3$ together contribute to the near-surface O$_3$ increases in winter during 1995–2019 associated with the changes in large-scale circulations. The anomalous atmospheric circulation is likely linked to the location of the midlatitude jet stream, which is influenced by ENSO cycle (Lin et al., 2015)."

* 342, I think it is a mistake to call the comparison of two five year periods "anomalous".

Response:
This is the difference between two 5-year averages. So we guess the "anomalous" could be used.

* 355-357, I think you got the signs of change wrong here. You showed decreasing in the summer (controls) and increasing in the winter (lessening titration).

Response:
    Yes, we have corrected it now.

* 355-357, You showed that it could only replicate the decreasing trend in the eastern summer and the increasing trend in the western winter. You showed that the trends for Eastern winter and western summer were *not* well captured. The trends were significantly different. So, it is wrong to say that it did well in the conclusion.

Response:
    We corrected the expression as follows: "This model can capture the $O_3$ decreasing trend over the EUS in summer and increasing trend over the WUS in winter during this time period, but largely overestimates the decreasing trend over WUS in summer and increasing trend over EUS in winter."

* 359-361, You need to be clear when you are talking about the model regions and when you are talking about the observed sites as sampled by the model. Are these trends at select sites? Are these trends representative of the larger region? Or the population weighted concentrations?

Response:
    We have clarified as that "In summer, our simulation results show that…".

* 364, This is a little less clear to me. The VOCs were also reduced. How do you distinguish between reduced VOC trends and reduced NOx OPE impacts on VOC trends.

Response:
    Because the biogenic emissions contribute the largest to the $O_3$ decreasing trend in summer, but they are kept constant during the simulation. Therefore, the $O_3$ decrease in summer is dominated by the reductions in $NO_x$ emissions.

* 391, This was a 3.7 ppb/decade decrease (not increase).

Response:
    Corrected.

* 392-393, The authors are offering only one of several equally permissible explanations. As you note, one is that the Asian trends are underestimated. Another is that the coarse model overestimates the decrease associated with

domestic reductions. Another is that stratospheric variability is underestimated. Another is that the trend in ships contributions are underestimated. What makes the authors confident that this is the only hypothesis to highlight?

Response:

    We agree many reasons can lead to the overestimate of the decreasing trend over WUS in summer by the model. Lin et al. (2017) found that the contribution from increasing Asian emissions offset that from the U.S. emission reductions, but our results showed the Asian contribution only offset a small amount of the domestic emission reductions. So we suspect that the "the Asian contribution to the $O_3$ trends in WUS is likely underestimated in this study". We did not say the overestimate of the decreasing trend over WUS is due to the bias in Asian contribution alone.

* 393-396, 1) The authors should show this model performance if their conclusions rely upon it. 2) The local peaks in China will depend on near surface vertical structure while the continental scale outflow may not. So, you could only say that it "consistent with the low contribution from Asian sources" since you cannot say that it definitively explains anything.

Response:

    To avoid misunderstanding, we have not removed this sentence from the manuscript.

* 396-397, It is unreasonable to think that model evaluation of China is worth discussing, while model evaluation over the US is not.

Response:

    Yes, we have now added model evaluation over the US. Please see the response above.

Reference:

Butler, T., Lupascu, A., Coates, J., and Zhu, a. S.: TOAST 1.0: Tropospheric Ozone Attribution of Sources with Tagging for CESM 1.2.2, Geosci. Model Dev, https://doi.org/10.5194/gmd-11-2825-2018, 2018.

Cooper, O. R., Gao, R.-S., Tarasick, D., Leblanc, T., and Sweeney, C.: Long term ozone trends at rural ozone monitoring sites across the United States, 1990-2010, J. Geophys. Res.: Atmospheres, 117, D22307, https://doi.org/10.1029/2012JD018261, 2012.

Emmons, L. K., Hess, P. G., Lamarque, J.-F., and Pfister, G. G.: Tagged ozone mechanism for MOZART-4, CAM-chem and other chemical transport models, Geosci. Model Dev., 5, 1531– 1542, https://doi.org/10.5194/gmd-

5-1531-2012, 2012.

Lamarque, J.-F., Emmons, L. K., Hess, P. G., Kinnison, D. E., Tilmes, S., Vitt, F., Heald, C. L., Holland, E. A., Lauritzen, P. H., Neu, J., Orlando, J. J., Rasch, P. J., and Tyndall, G. K.: CAM-chem: description and evaluation of interactive atmospheric chemistry in the Community Earth System Model, Geosci. Model Dev., 5, 369–411, https://doi.org/10.5194/gmd-5-369-2012, 2012.

Lin, M., Fiore, A. M., Cooper, O. R., Horowitz, L. W., Langford, A. O., Levy, H., Johnson, B. J., Naik, V., Oltmans, S. J., and Senff, C. J.: Springtime high surface ozone events over the western United States: Quantifying the role of stratospheric intrusions, J. Geophys. Res. Atmos., 117, https://doi.org/10.1029/2012JD018151, 2012.

Lin, M., Horowitz, L. W., Payton, R., Fiore, A. M., and Tonnesen, G. S.: US surface ozone trends and extremes from 1980 to 2014: quantifying the roles of rising Asian emissions, domestic controls, wildfires, and climate, Atmospheric Chemistry and Physics, 17, 2943–2970, https://doi.org/10.5194/acp-17-2943-2017, 2017.

Price, C., Penner, J., and Prather, M.: NOx from lightning 1, Global distribution based on lightning physics, J. Geophys. Res., 102, 5929–5941, https://doi.org/10.1029/96JD03504, 1997.

Szopa, S., V. Naik, B. Adhikary, P. Artaxo, T. Berntsen, W.D. Collins, S. Fuzzi, L. Gallardo, A. Kiendler-Scharr, Z. Klimont, H. Liao, N. Unger, and P. Zanis: Short-Lived Climate Forcers. In Climate Change 2021: The Physical Science Basis. Contribution of Working Group I to the Sixth Assessment Report of the Intergovernmental Panel on Climate Change [Masson-Delmotte, V., P . Zhai, A. Pirani, S.L. Connors, C. Péan, S. Berger, N. Caud, Y. Chen, L. Goldfarb, M.I. Gomis, M. Huang, K.  Leitzell, E.  Lonnoy, J.B.R.  Matthews, T. K.  Maycock, T. Waterfield, O. Yelekçi, R. Yu, and B. Zhou (eds.)]. Cambridge University Press, Cambridge, United Kingdom and New York, NY, USA, pp. 817–922, https://doi.org/10.1017/9781009157896.008, 2021.

Tilmes, S., Lamarque, J. F., Emmons, L. K., Kinnison, D. E., Ma, P. L., Liu, X., Ghan, S., Bardeen, C., Arnold, S., Deeter, M., Vitt, F., Ryerson, T., Elkins, J. W., Moore, F., Spackman, J. R., and Val Martin, M.: Description and evaluation of tropospheric chemistry and aerosols in the Community Earth System Model (CESM1.2), Geosci. Model Dev., 8, 1395-1426, https://doi.org/10.5194/gmd-8-1395-2015, 2015.

Wesely, M. L.: Parameterizations for surface resistance to gaseous dry deposition in regional-scale numerical models, Atmos. Environ., 23, 1293–1304, https://doi.org/10.1016/0004-6981(89)90153-4, 1989.

---

## Author Comment (AC2)

**Manuscript # acp-2022-678**

**Responses to Referee #2**

The manuscript of Li et al. investigates trends of ground-level over the US for 1995 – 2019. They use a source attribution method (tagging) to attribute the ozone trends to trends of emissions of VOCs and NOx from different sectors and regions.

The topic of the manuscript is very interesting and it fits into the scope of ACP. However, the manuscript needs major revisions before it can be reconsidered for ACP.

General remarks:

1) The authors report that during winter time ozone increases due to NOx titration. They use, however, a rather coarse resolved global model. It is well known that at this resolution NOx titration is usually underestimated by the model. Therefore, I agree with referee #1 that a more detailed model evaluation is needed. The authors should also discuss how the ability of the model to capture the observed trends only partly (e.g. increasing trend in EUS in winter is strongly overestimated) influence the conclusions. Further, it would be interesting to analyse if the model is able to capture the chemical regimes in EUS and WUS correctly.

Response:
     Thank you for the suggestion. We have now added the distribution of observed and modeled surface $O_3$ in the United States for summer and winter in 1995 and 2019 in Figure S4. We have also included a model evaluation section as:
     "Figure S4 compares the simulated near-surface $O_3$ concentrations with those from observations in 1995 and 2019, respectively. In general, the model overestimates $O_3$ concentrations in the U.S. in both summer and winter by 10–40%. It can capture the $O_3$ seasonality that high concentrations in summer and low concentrations in winter. The spatial distributions can also be roughly captured by the model, with statistically significant correlation coefficients between simulations and observations in the range of 0.21–0.45. From 1995 to 2019, the $O_3$ concentrations in the U.S. decreased in summer and increased in winter presented in observations. The model can produce the sign of the changes, but has large biases in magnitudes, which will be discussed in the following section."
     The overestimation of $O_3$ trend in western U.S. in winter could be attributed to the overestimation of the weakened $NO_x$ titration, related to the coarse model resolution. We have now added a note that "The model also tends to

overestimate the weakening of NOx titration in winter, leading to the biases."

[Figure]

**Figure S4.** The simulated (contours) and observed (scatters) seasonal mean near-surface O₃ concentrations over the United States in JJA (left) and DJF (right) and in 1995 (top) and 2019 (bottom). The correlation coefficient and normalized mean bias (NMB, ∑ (Model − Observation) / ∑ Observation× 100%) are shown on top right of each panel.

As to how the ability of the model to capture the observed trends only partly influence the conclusions, we have now added sentence as follows: "Lin et al. (2017) found that the contribution from increasing Asian emissions offset that from the U.S. emission reductions, resulting in a weak O₃ trend in WUS. In this study, the Asian NOx emissions only contribute to 0.6 ppb/decade of the total positive trend in WUS in summer, much lower than the 3.7 ppb/decade decrease attributable to the domestic emission reductions, suggesting that the Asian contribution to the O₃ trends in WUS is likely underestimated in this study. The bias of O₃ simulation in China may also lead to a bias in the wintertime O₃ trend over EUS. Additionally, international shipping can have a disproportionately high influence on tropospheric O₃ due to the dispersed nature of NOx emissions (Butler et al., 2020; Kasibhatla et al., 2000; von Glasow et al., 2003), together with the weakened NOx titration, resulting in the overestimation of O₃ trends. The fixed CH₄ concentration during simulations also biased the modeled O₃ trends in this study. The coarse model resolution also contributed to the biases. The overestimate of O₃ trend over EUS in winter, likely related to the bias in NOx titration, implies the overestimate of source contributions to the trends in magnitude."

Although determination of the chemical regime is typically made according to the indicator ratio P_{H2O2} / P_{HNO3} (the ratio between the production rates of

hydrogen peroxide and nitric acid), our simulations did not output these two variables. Alternatively, we show here the ratio of $NO_x$ to NMVOCs emissions to support our results as the following: "Note that, during 1995–2019, the molar ratio (mol N /mol C) of emitted $NO_x$ to NMVOCs reduced from 0.11 to 0.07 in the WUS and from 0.14 to 0.07 in the EUS, confirming the enhanced $NO_x$-sensitive condition during the analyzed time period."

2) Even though the CEDS emissions are well documented by Hosely et al., 2018 the authors should discuss these emissions in more detail as the results of the study heavily depend on the emission inventory. How do the trends in the emissions of Hosley et al., 2018 for example compare to the trends in McDuffie et al. 2020, the CAMS or the EDGAR emissions? What is the influence of the inconsistency in the aviation emissions in CEDS (Thor et al., 2022) on the results?

Response:
    Thank you for the suggestion, we have added the following discussion:
    "As the results of the study heavily depend on the emission inventory, here the potential bias in emissions are also discussed. Compared with the previous CEDS version used in this study (hereafter CEDS$_{Hoesly}$), the updated CEDS inventory (hereafter CEDS$_{GBD-MAPS}$) (McDuffie et al., 2020) incorporates updated activity data. For $NO_x$, the global emission from CEDS$_{GBD-MAPS}$ is smaller than that of CEDS$_{Hoesly}$ after 2006 and shows a fast decreasing trend. By 2014, global emission of $NO_x$ is about 10 % lower than the CEDS$_{Hoesly}$ estimate. These differences are mainly reflected in the industrial and residential sectors in China, followed by the transportation sector in India and Africa. For global emission of NMVOCs, which remains relatively unchanged between the CEDS$_{Hoesly}$ and CEDS$_{GBD-MAPS}$ inventories (Fig. 6 in McDuffie et al. 2020). The global $NO_x$ emission from EDGAR v4.3.2 inventory is less than CEDS$_{Hoesly}$ (Crippa et al., 2018). This difference in $NO_x$ emissions may reduce $O_3$ trends in U.S. from foreign contributions, especially from East Asia. Recent study also reported a difference in $NO_x$ emission distribution between CMIP5 and CMIP6 related to an error in data pre-processing in CEDS, leading to a northward shift of $O_3$ burden in CMIP6 (Thor et al., 2023). The aviation emissions should be corrected in future studies of $O_3$ simulations."

3) The labels and fonts in many figures are too small. All of the labels/fonts (and also the station symbols in Fig 2.) needs to be enlarged.

Response:
    Thank for you suggestion. We have made corresponding adjustments.

4) The model description misses a lot of basic information. Even though the CAM4-chem model is well known, most important information should be given

in the manuscript. As example detailed information about the chemical mechanism is missing. How is dry and wet deposition represented? What variables are nudged? Further I am missing information about the emission totals for natural emissions (lightning NOx, biogenic VOCs and soil NOx) as well as the global emissions. All of these information are important to compare results from different studies with each other. Of course not all of these things need to be discussed in detail. Some are also fine in the supplement (for example detailed information about the emissions).

Response:
  Thank you for the suggestion. We have now revised the text as follows:
  "The model configuration uses a comprehensive tropospheric chemistry mechanism based on the Model for Ozone and Related chemical Tracers version 4 (MOZART-4) (Emmons et al., 2010, 2012). Model configurations simulate wet deposition of gas species using the Neu and Prather (2012) scheme. Dry deposition is represented following the resistance approach originally described in Wesely (1989). Stratosphere-troposphere exchange of $O_3$ is treated by setting $O_3$ to stratospheric values as their climatological means over 1996–2005 at the tropopause (Lamarque et al., 2012), which is affected by atmospheric circulation and experiences the same loss rates as $O_3$ in the troposphere (Tilmes et al., 2016)… The zonal and meridional wind fields are nudged towards the MERRA-2 reanalysis."
  In addition, we have provided the details of the information of the total global emissions in the supplement.

**Table S2.** Global total emissions of $NO_x$, NMVOCs and CO for different sectors in 1995 and 2019.

| | year | AGR | ENE | IND | RCO | SHP | TRA | SLV | WST | BMB | SOIL/BIO | AIR |
|---|---|---|---|---|---|---|---|---|---|---|---|---|
| $NO_x$ | 1995 | 1.23 | 9.19 | 4.24 | 3.16 | 5.11 | 11.34 | | 0.39 | 4.48 | 7.98 | 0.67 |
| TgN/yr | 2019 | 1.61 | 7.83 | 5.25 | 2.53 | 6.02 | 11.26 | | 0.74 | 4.00 | 7.98 | 1.19 |
| NMVOCs | 1995 | 4.86 | 24.99 | 8.27 | 31.16 | 2.59 | 35.97 | 23.32 | 2.70 | 64.28 | 664.87 | |
| TgC/yr | 2019 | 7.60 | 35.51 | 11.63 | 28.93 | 3.15 | 25.30 | 31.90 | 2.76 | 61.99 | 664.87 | |
| CO | 1995 | | 15.97 | 42.42 | 113.09 | 0.21 | 102.52 | | 3.81 | | 68.51 | 0.19 |
| TgC/yr | 2019 | | 28.17 | 40.41 | 97.63 | 0.33 | 55.63 | | 7.25 | | 68.51 | 0.31 |

5) The authors need to clarify the units they use. They use ppb as unit which sounds like (volume/mass?) mixing ratios but use the term concentration throughout the manuscript. Please clarify if you consider mixing ratios or concentrations. Also, for the emissions totals the authors should specify what NOx and NMVOCs are (see below for more details).

Response:
    We have now clarified parts per billion (ppb, volume ratio in this study) and described $NO_x$ and NMVOC with TG N and TG C, respectively. The mixing ratio is sometimes expressed as concentration in many studies, so we prefer to keep it as it is.

6) In the model simulations CH4 mixing ratios are kept fix at 1750 ppb. This represents ~ 1990 levels (https://gml.noaa.gov/ccgg/trends_ch4/). Until 2019 CH4 levels have been increased to ~ 1880 ppb which is a increase of ~7--8 %. This increase influences ozone production and Butler et al., 2020 found very inhomogeneous changes of the contributions by CH4 increases. Therefore, I suggest to perform an additional simulation with an CH4 increase.

Response:
    This is a good suggestion to quantify the contribution of increasing $CH_4$ to $O_3$ trend. However, it takes several months for one long-term simulation, which beyond our current computational resource. We have added a note that "It is noticed that the fixed $CH_4$ mixing ratio during simulations also biased the modeled $O_3$ trends in this study, which deserves further investigation with the varying $CH_4$ levels in future studies."

7) The authors should reconsider the choice of tagging labels. In my opinion the region North America should have been spitted into US and Canada. Further, important information are lost because the shipping emissions have the ROW tag in the "region tagging" runs. Why are they not tagged as shipping/oceanic in the "regional tagging" runs? Further, I wonder why the results of many "unchanged" sectors changed between the "sector" and "region" tagging runs. Shouldn't the results for the tags "STR", "LGT", "AIR", "SOIL" be identical in Fig 5. and 7? If only anthropogenic sources get either sector or region tags the results of the natural sources should not change? In my view this is a very critical inconsistency which needs to be clarified (maybe I also don't understand the approach correctly). In addition, the authors should motivate the special category for CO in more detail. Many information are lost by lumping all CO emissions in one category.

Response:
    For the choice of tagging labels, we referred to the HTAP Tier 2 receptor regions. As to the ROW tag in the "region tagging" runs, according to Fig. 1, which includes emissions from the Arctic and Antarctic as well as southern Africa, Oceania, and northern Asia, not just the oceans. We would like to add more tagging labels, but the tagging system crashed when more tags were added, which should be addressed in the next version of the code.
    Thanks for pointing the tag issue, the results for the tags "STR", "LGT",

"AIR", and "SOIL" should be similar between sector and regional runs. The larger discrepancies for these tags in last version are due to an error in the division of emissions used in our model and it has been corrected now.

For the CO tagging, we have clarified as "CO also has a longer lifetime and lower reactivity than most NMVOCs, separately tagging of CO is more conducive to distinguish its contribution to $O_3$ from other NMVOCs. Therefore, the lumped total CO is separately tagged in the sector attribution simulations, but the CO is not specifically tagged in the regional attribution simulations due to the computational limit."

8) The manuscript lacks a detailed discussion of the results in comparison to other global source attribution studies. Are the results in accordance with other studies? For example are results of O3 from STR, lightning or biogenic sources comparable with other studies? Much more comparison with previous work is needed (see below for some references; in addition also Guo et al. 2017 .and all the studies from the HTAP framework (https://htap.org/) can be of interest here). In my opinion also the introduction needs to be improved (see detailed comments below).

Response:

Thank you for your suggestion. We corrected statements and added tables to describe overall trends.

We have now compared the results of our simulations (Figs. S7–S9) with Butler et al. (2018, 2020) and added the following in the discussion: "Compared with Butler et al. (2018), the simulation in this study shares similar source sector contributions to the zonal average of $O_3$ concentrations at the surface and 400 hPa in 2010 (Figs. S7 and S8 in this study and Figs. 5 and 6 in Butler et al. (2018)). The contributions from the stratosphere and lightning $NO_x$ are relatively higher in this study than Butler et al. (2018). This may be related to the different anthropogenic emission inventories used, causing different $O_3$ production/loss efficiencies by natural precursors. When comparing the contributions from different source regions to surface $O_3$ concentrations in North America, $NO_x$ emissions from East Asia, South Asia, North America, and Europe contributed 2.2, 1.1, 8.3, and 0.7 ppb of the surface $O_3$ in North America, respectively (Fig. S9) in this study, which are also similar to those from Fig. 4 in Butler et al. (2020). Both studies show the contributions of anthropogenic NMVOCs to surface $O_3$ concentrations in North America are less than 10 ppb."

9) I am wondering about the trend of O3 due to aircraft emissions. Usually most aircraft emissions take place in the (upper) troposphere and not near ground-level. Therefore I wonder if there is a trend of O3 from aviation (check values in the upper troposphere) or if there is an increase in downward transport. If so, it would be interesting to separate effects due to increased emissions and due to changes in dynamics.

Response:
We have now added a figure and the following to illustrate the O₃ trend contributed by aircraft emissions as "Although aircraft NOₓ emissions slightly increased, but O3 attributed to aircraft NOx emissions shows positive trends as large as 0.4±0.0 and 0.6±0.0 ppb/decade in WUS and EUS, respectively, because aircraft emissions are injected directly into the upper troposphere and lower stratosphere in a low ambient NOₓ condition and have a much higher O₃ enhancement efficiency than surface emissions (Hodnebrog et al., 2011). It can be confirmed that the NOₓ from aircraft contributes to the increase in O₃ concentrations at 250 hPa in high latitude regions of the Northern Hemisphere during 1995–2019 (Fig. S6)."

[Figure]

**Figure S6.** Annual O₃ trends contributed by aircraft at 250hPa from 1995-2019.

Detailed comments:

p4l70-p4l87: This section needs some corrections. The perturbation approach and labeling techniques are two different methods answering different scientific questions. The perturbation approach provides (potential) impacts. Tagging provides contributions. There is many literature discussing this issue which can be checked for more details– some are: Grewe et al. 2010, Emmons et al. 2012, Clappier et al. 2017 and Tunis et al., 2020.

Response:
We have now corrected the description as "One method of obtaining an O₃

source-receptor relationship is to zero out or perturb emissions from a given source region or sector in sensitivity simulations along with a baseline simulation, which gives information about the response of O₃ to changes in precursor emissions." "The tagging approach produces information about the contribution of precursor emissions to the total amount of O₃ (Butler et al., 2020). The perturbation and tagging methods are two different methods answering different scientific questions, with the first for the impacts and the last for the contributions (Grewe et al. 2010, Emmons et al. 2012, Clappier et al. 2017 and Thunis et al., 2019)."

p4l91: This is not correct. There are approaches applied on the regional scale which use chemical indicators (Dunker et al., 2012, Kwok et al., 2015). However, there are also approaches on the regional scale which do not use chemical indicators (e.g. Lupaşcu and Butler, 2019; Mertens et al. 2020)

Response:
    Thanks for the suggestion. We revised the description to "In some regional models, O₃ apportionment is based on the ratio of chemical indicators to determine the regime of O₃ generation (e.g., VOC-limited or NOₓ-limited regimes) and then attribute the generation of O₃ to the tag carried by a certain precursor (VOCs or NOₓ), which however cannot simultaneously attribute O₃ production to NOₓ and VOCs, respectively (Dunker et al., 2002; Kwok et al., 2015), while some models do not use the chemical indicators (Lupaşcu and Butler, 2019; Mertens et al., 2020)."

P5l97: I think this heavily depends on how the boundary conditions are implemented (see literature above).

Response:
    Here we are trying to say the regional model can not separate the regional contributions from several source regions outside the domain. We have now clarified the description.

P5l100ff: There are also global approaches which use a sector wise attribution or a combination of sector wise and regional attribution (Emmons et al. 2012, Grewe et al. 2017, Butler et al. 2018). Butler et al. 2018 includes a comprehensive overview of different approaches which the authors could check.

Response:
    Yes. Here we would like to highlight that some global models directly tag the O₃ production rather than the precursor emissions. These literatures have been cited in the manuscript.

P6L120: Is O3 is nudged at the tropopause towards 'stratospheric' values?

What happens with the stratosphere tagged tracer?

Response:

We have revised the description to "Stratosphere-troposphere exchange of $O_3$ is treated by setting $O_3$ to stratospheric values as their climatological means over 1996–2005 at the tropopause (Lamarque et al., 2012), which is affected by atmospheric circulation and experiences the same loss rates as $O_3$ in the troposphere (Tilmes et al., 2016)."

The STR tag is neither from $NO_x$ nor VOCs emissions. In both $NO_x$ and VOCs tagging, initial conditions for $O_x$ species in the stratosphere were tagged with STR. In addition, the photolysis of $O_2$ and $N_2O$ ultimately produces the $O_3$, which is all tagged as STR. They can be transported downward via atmospheric circulation and contribute to the near-surface $O_3$ concentrations.

P7I152ff: See also general comments above. Why not tagging shipping emissions separately as Ocean (see Butler et al., 2020). What about aviation in this list. If I understand the analysis correctly aviation has been tagged as sector in the regional runs?

Response:

"ROW" includes emissions from the Arctic and Antarctic as well as southern Africa, Oceania, and northern Asia in addition to oceans. Due to the limitation of the number of tagging (causing model crash), we merged shipping emission into ROW tag.

Yes, aviation has been tagged as sector in the regional runs. Because aircraft emissions covers both land and ocean regions, it is not so reasonable to determine the regional contribution of aviation emissions only by the emissions over the region.

P7I164: See also general remarks. Is this the explanation why CO is lumped?

Response:

We have clarified as "CO also has a longer lifetime and lower reactivity than most NMVOCs, separately tagging of CO is more conducive to distinguish its contribution to $O_3$ from other NMVOCs. Therefore, the lumped total CO is separately tagged in the sector attribution simulations, but the CO is not specifically tagged in the regional attribution simulations due to the computational limit."

p7I168: What about emissions of SO2 and NH3?

Response:

They are also from CEDS. Now added.

P8l177: Please specify the lightning NOx total emissions?

Response:
    Specified as "Lightning emissions of $NO_x$ are estimated using online parameterization based on simulated cloud top heights from Price et al. (1997), which is scaled to provide a global annual emission of 3–5 Tg N $yr^{-1}$ as Lamarque et. al. (2012)."

P9l208-p9l225: See also general remarks. For summer EUS many stations show an increase of O3. It seems that this is not captured by the model? For winter EUS stations some stations show no or even a decreasing trend. These trends are not captured by the model. Please comment.

Response:
    Yes, in summer some studies found $O_3$ increased in Los Angeles and some sites in the central United States, decreased in Nevada and Utah (Cooper et al., 2012; Lin et al., 2017). The coarse grid resolution of the global model cannot accurately capture the differences between cities or sites, but the relatively stronger $O_3$ reduction trend in the central WUS region (Nevada and Utah) is still well reproduced on the regional scale. It is the same reason for winter. We have added the explanation in the manuscript.

Fig 3 : Why do VOC emissions of the ENE sector increase while NOx emissions decrease? Please comment and compare the trend of ENE with other emission inventories. Please specify if emissions are Tg(N), TG(NO) etc. (also for VOCs)

Response:
    Thank you for your suggestion. We now specify $NO_x$ as TG N and VOC as TG C and modify the corresponding figures.
    Over the past 10–20 years in the US, the reduction in coal-fired power plant emissions have resulted in emission reductions in $NO_x$ (Krotkov et al., 2016; Duncan et al., 2013; Castellanos and Boersma, 2012; de Gouw et al., 2014). Over this same time period, however, oil and gas production in key regions in the US has more than tripled between 2007 and 2017 (EIA, 2020), which resulted in VOC emissions increasing.

P10l235: Fig 4 and 5 should be reordered; same for 6 and 7.

Response:
    Reordered.

P10l246: A more detailed analysis of the change of the O3 production efficiency would be very valuable here.

Response:

Thanks for your suggestion, we have added the following: "Note that, during 1995–2019, the molar ratio (mol N /mol C) of emitted $NO_x$ to NMVOCs reduced from 0.11 to 0.07 in the WUS and from 0.14 to 0.07 in the EUS, confirming the enhanced $NO_x$-sensitive condition during the analyzed time period."

P10l247: See also general remarks: Information about the trends of global emissions (e.g. shipping etc.) would be very valuable here.

Response:

We have added a figure below to describe the trend of shipping emissions and $O_3$ contribution, then modified the shipping-related part as follows:

In recent decades, emissions from international shipping have increased rapidly (Eyring, 2005; Müller-Casseres et al., 2021), but have declined near the coast of the United States. Due to a strong chemical sink associated with photolysis of $O_3$ with subsequent production of hydroxyl radical (OH) from water vapor in summer (Johnson et al., 1999), the effect of increased emissions of the far-shore ocean on the continental United States was blunted. But the increase in shipping emissions inland tends to increase $O_3$ concentrations in eastern U.S.

In winter, the decrease in near-shore shipping weakened the $NO_x$ titration, together with the weakened $O_3$ chemical sink from water vapor in winter, leading to large increasing trends of $O_3$ by 0.8±0.1 and 1.0±0.1 ppb/decade, respectively, in the WUS and EUS during 1995–2019.

[Figure]

**Figure S5.** Trends of shipping emissions of $NO_x$ and $O_3$ trends contributed by shipping emissions in JJA and DJF from 1995 to 2019.

P10L257ff: Could you please explain the argumentation here in more detailed? I think additional analysis would help here to make this point more clear.

Response:
 $O_3$ decreases in summer over the U.S. due to the reductions in $NO_x$ emissions. At the same time, the VOCs do now have significant decrease, that leads to a decrease in $O_3$ production efficiency by VOCs. We have revised the sentence as follows: "This does not actually mean that $CH_4$ and biogenic NMVOCs themselves contributed to the overall $O_3$ trend through changing the precursor levels since they were kept constant during simulations; rather, mainly due to the reductions in $NO_x$ emissions, $O_3$ production efficiency by reactive carbon species decreases, leading to decreasing trends of $O_3$ contribution by $CH_4$ and biogenic NMVOCs."

P11l278ff: This sentence is very long. I suggest to split it up.

Response:
 Changed.

P12l295: See general remarks. Why did you applied one combined tag for North America and not separate tags for US and Canada?

Response:
 The choice of tagging labels we refered to HTAP Tier 2 receptor regions and the tagging system will crash when more tags are added.

P13l318: I don't understand the sentence. Please explain. Thanks!

Response:
 The combined natural source means LGT, STR and SOIL. In winter, with the reductions in domestic anthropogenic $NO_x$ emissions, the weakened $NO_x$ titration process leads to an increase in the $O_3$ production efficiency from these sources.

P13L334: The figure suggest that none of the results are significant (by the way; with which method did you check for significance. Please explain in detail were appropriate).

Response:
 Based on our MET experiments, it is significant in the central U.S. in winter (Fig. 8b). We calculate linear least-squares regressions for $O_3$ and years,

considering the P<0.05 to be significant.

Section 3.4: See general comments above. More details/analysis are missing here. In my opinion especially an analysis of the contributions would be very valuable here. How large is for example the trend of ozone from the stratosphere due to changes in the dynamics.

Response:
We supplement the trend of O3 contribution from each emission source to the U.S. in winter in Fig. 8 and added more details as follows: "Variations in the circulation facilitate O3 transport from upper altitudes to the surface, as well as foreign contributions from Asia, which is consistent with the finding in Lin et al. (2015). The O3 increasing trend in winter over the U.S. attributing to stratospheric injection and Asian NOₓ emissions due to dynamics are both 0.2±0.1 ppb/decade (Fig. 8e). Therefore, changes in anthropogenic emissions are the main factor affecting O3 trends."

[Figure]

**Figure 8.** Linear trends (ppb/decade) of simulated (a) JJA and (b) DJF mean near-surface O3 concentrations during 1995–2019. Differences between the

first (1995–1999) and last (2015–2019) five years during 1995–2019 (last–first) in DJF mean (c) 850 hPa horizontal winds and (d) meridional winds and vertical velocity averaged over 90–105°W. Areas without hatches in (a) and (b) and red arrows in (c) and (d) indicate statistical significance with 95% confidence. All results are from the MET experiments. Linear trends (ppb/decade) of near-surface $O_3$ concentrations in winter over the U.S, contributed by the NOx (e, g) and reactive carbon (f, h) emissions from various source sectors (e, f) and regions (g, h). The increasing and decreasing trends marked with red and blue color numbers, respectively, indicate statistical significance with 95% confidence. Contributions from source regions EAS, SAS and SEA are combined to ASIA. Some sources having small contributions are combined and shown as OTH.

P14l355: I don't agree with the statement that the model captures the trends 'well'. Please rephrase.

Response:
   We change the expression to "This model can capture the $O_3$ decreasing trend over the EUS in summer and increasing trend over the WUS in winter during this time period, but largely overestimates the decreasing trend over WUS in summer and increasing trend over EUS in winter."

P15l374ff: What is the reason for the increase of shipping and aviation – emissions or dynamics? Please analyse in more detail.

Response:
   According to the newly added Fig. 8e and Fig. 5, the $O_3$ increase attributing to shipping and aviation $NO_x$ emissions due to dynamics are 0.07 and 0.05 ppb/decade, respectively, much lower than 0.76/0.99 and 0.43/0.55 ppb/decade in the default simulation. Therefore, changes in emissions are the main factor.

Reference:

[revised manuscript text omitted]

---

## Author Response (AR2)

**Responses to Referee #1**
**Summary**

The authors have thoughtfully responded to my requests. The model performance results really highlights the currently poor performance. Although the performance does not inspire confidence, the authors have clearly documented the performance and future readers may make up their own minds. I do have a few minor recommendations that do not require subsequent review.

The model performance section should be improved a bit before publication. As written, the performance figure is in the supplement, but it should be moved to the main body. The authors statement that it "can capture the O3 seasonality" is not particularly compelling. The R values from 0.21 to 0.45 are also not particularly good. That being said, the authors are currently comparing sites to a coarse grid. Obviously, any sub grid scale variability will not be captured. The authors could consider some sort of monitor averaging to account for this before calculating R.

Response:
      Thank you for the suggestions. We have moved the performance figure to the main body. Both the model and observations show high values in summer and low values in winter. This pattern can be captured by the model. We have revised the description as "It can capture the seasonal pattern of $O_3$ that high mixing ratios in summer and low mixing ratios in winter." We agree with the review that monitor averaging method could be better, but is unlikely to largely improve the statistics.

The authors also seem to have moved their primary result of trends to a supplemental table (Table S1). I hope that was a mistake as the values should be presented in the main body.

Response:
      We have moved it to the main body.

With those very minor updates, I support publication of these results. I'll note, however, that these results largely call for higher resolution simulations or a focus on metrics (e.g., MDA8 or MDA1) that the model simulates better.

Response:
      Thank you. Since the output of the current simulation results is monthly average data, we are unable to calculate MDA8 or MDA1. We will take your

suggestion into consideration in our future studies with higher model spatial and temporal resolutions.

**Line-by-Line**

171: We [do] not

Response:
    We have revised it.

Observation methods:
The new figure S4 shows the observations sites and highlights that the observation method section needs a bit of work. Right now, you say that each site must have

Response:
    We guess the reviewer mean the sentence for the site selection. We have modified the description as "Seasonal mean for any site that has less than 50% data availability in any month of a season is discarded following Lin et al. (2017). $O_3$ trends is calculated only when the seasonal data availability is greater than 85% during the analyzed period (more than 22 years)."

**Manuscript # acp-2022-678**

**Responses to Referee #2**

The authors did good job with their detailed repose to my comments/questions. The manuscript, however, needs (at least) minor revisions. Most of the additional analyses and information were just put to the conclusion & discussion section or the Supplement. In addition, the changes in the manuscript need a more detailed proof-reading, because it is very hard to follow some of the newly added paragraphs. Finally, at least to my opinion, the revised manuscript is partly very confusing because it jumps between (sub)-figures.

We thank the reviewer for all the insightful comments. We have now made detailed proof-reading throughout the text and reordered and referenced all figures in the manuscript. Please see our point-by-point response below.

Please find my detailed points below:

1) The order of figures is not the same order as they are referenced in the manuscript; this does also apply for the Supplement. Please also check if all supplementary data is referenced in the manuscript. For example, a reference to Table 2 seems to be missing.

Response:
    Thank you for the suggestion. We have now made corresponding adjustments to all figures and tables in the main text and the supplement.
    For example:
    "The time series of the source contributions from $NO_x$ and reactive carbon emissions are shown in Fig. 5 and the $O_3$ trends in the U.S. attributed to different emission source sectors are shown in Fig. 6."
    "Time series of the source contributions are shown in Fig. 7 and the $O_3$ trends in the U.S. attributed to different emission source regions are presented in Fig. 8."
    Table 1 (the original Table 2) has now been mentioned in the *Emissions and Observations* section.

2) To my opinion Table S1/Fig. S4 are an essential part of the model evaluation and should be part of the manuscript. Do you agree?

Response:
    Agree. They have been moved into the manuscript.

3) The discussion about the trends of the emissions is placed at the end of the

manuscript. To my opinion this discussion would fit much better to the parts where the emissions/Fig 3 are discussed. In addition, I am missing a discussion about how differences in the emissions might affect the results of the study. Are your estimates of the trends from domestic/Asian emissions are likely at the upper or lower end (given differences in the emissions).

Response:
  Thank you for the suggestion. We have modified the emission description and moved it to the *Emissions and Observation* at the *Methods* section, as following:

  "Many studies have reported that the previous CEDS version 20160726 (hereafter $CEDS_{2016}$) has large biases in the regional emission estimates (e.g., Cheng et al., 2021; Fan et al., 2018). In this study, the CEDS version 20210205 is used (hereafter $CEDS_{2021}$), which builds on the extension of the CEDS system described in McDuffie et al. (2020) and extends the anthropogenic emissions to year 2019. It updates country-level emission inventories for North America, Europe and China and has considered the significant emission reductions in China since the clean air actions in recent years. The global total $NO_x$ emission from $CEDS_{2021}$ is lower than that of $CEDS_{2016}$ after 2006 and it shows a fast decline since then. In 2014, the global total anthropogenic emission of $NO_x$ in $CEDS_{2021}$ is about 10% lower than the $CEDS_{2016}$ estimate. This difference is mainly reflected in the $NO_x$ emissions in China and India. $CEDS_{2021}$ has a lower estimate of the global NMVOCs emission than $CEDS_{2016}$ by more than 10% during the recent decades, attributed to lower emissions from Africa, Central and South America, the Middle East and India. The using of the $CEDS_{2021}$ emission inventory in this study could reduce the contributions of $NO_x$ emissions from East Asia and South Asia to the U.S. $O_3$ mixing ratios and trends, as compared to $CEDS_{2016}$. However, recent study reported a difference in aviation emission distribution of $NO_x$ between CMIP5 and CMIP6 related to an error in data pre-processing in CEDS, leading to a northward shift of $O_3$ burden in CMIP6 (Thor et al., 2023). Therefore, the contribution of the aircraft emissions of $NO_x$ to the $O_3$ mixing ratios could be overestimated at high latitude regions."

4) Further, I am missing a discussion on how the model biases might affect the derived trends. In l229ff the authors write "..which will be discussed in the following section" but I am missing this discussion.

Response:
  Here, "The model can produce the sign of the changes, but has large biases in magnitudes, which will be discussed in the following section." Then we describe the biases in magnitudes between observation and simulation based on Table 1. The discussion of model biases is mainly in the conclusions and discussions as the following:

"Compared to observations, the decreasing trend of $O_3$ mixing ratios over WUS in summer and increasing trend over EUS in winter are overestimated in the CAM4-chem model. Because most $O_3$ monitors are located in urban areas and these areas generate strong $O_3$ during the day and have strong oxidation titration at night, the daily and grid averaged $O_3$ mixing ratios output by the model could be inconsistent with the urban observations. The overestimate of $O_3$ trend in the EUS might be related to a potential biased model representation of vertical mixing in winter. Large uncertainties existing in the emissions also result in the biases in the $O_3$ simulation. Lin et al. (2017) found that the contribution from increasing Asian emissions offset that from the U.S. emission reductions, resulting in a weak $O_3$ trend in WUS. In this study, the Asian $NO_x$ emissions only contribute to 0.6 ppb/decade of the total positive trend in WUS in summer, much lower than the 3.7 ppb/decade decrease attributable to the domestic emission reductions, suggesting that the Asian contribution to the $O_3$ trends in WUS is possibly underestimated in this study. We also found that the model did not capture the significant increase in summertime $O_3$ levels in China in recent years, which could explain the low contribution from Asian sources. Additionally, international shipping can have a disproportionately high influence on tropospheric $O_3$ due to the dispersed nature of $NO_x$ emissions (Butler et al., 2020; Kasibhatla et al., 2000; von Glasow et al., 2003), together with the weakened $NO_x$ titration, resulting in the overestimation of $O_3$ trends. The fixed $CH_4$ mixing ratio during simulations also biased the modeled $O_3$ trends, which deserves further investigation with the varying $CH_4$ levels in future studies. The coarse model resolution also contributed to the biases. The overestimate of $O_3$ trend over EUS in winter, likely related to the bias in $NO_x$ titration, implies the overestimate of source contributions to the trends in magnitude."

5) I suggest that information on how you calculated the trends (least square linear trend) and your definition of "significance" should be an essential part of the method section.

Response:
    Thank you for the suggestion. We have now added following sentences in the *Emissions and Observation* section: "Trends in this study are calculated based on the linear least-squares regressions and the significance is identified through the F test with the 95% confidence level."

6) The authors replied that "the results for the tags "STR", "LGT", "AIR", and "SOIL" should be similar between sector and regional run".

I don't understand this answer. If there are two simulations – one with regional and one with sectoral attribution – with the same atmospheric dynamics and the same emissions, the contributions of identical tags (air, str etc.) should be the same between the two simulations? As example, the trend of STR in Fig 5e

is 0.64 ppb/decade and the trend of STR in Fig. 7e is 0.70 ppb/decade.
I would expect that the two tags show the same trend in both runs. Most likely I get something wrong here. Could you please explain this in more detail? Is the atmospheric dynamics different? This should also be explained in the manuscript.

Response:
    The differences between the sector and regional simulations could be due to the slight difference in the atmospheric dynamics related to the nudging of the wind fields. The zonal and meridional wind fields are nudged to the reanalysis data at a 6-hourly relaxation timescale, rather than completely driven by the reanalysis data as in the chemical transport models. Also, other meteorological factors like temperature and humidity were not nudged to the reanalysis. However, the slight differences would not affect the main conclusion of this study. We have added the explanation in the manuscript.

7) The changed Sect 3.4 jumps between the subfigures (8a, b followed by 8e). Some information are doubled (trends stratosphere). Please rephrase the section completely and think about splitting Fig 8 into two figures."

Response:
    We have now split the Fig. 8 into two figures and rephrased the section as the following:
    "Many studies have reported that $O_3$ spatial distribution is strongly modulated by changes in large-scale circulations (e.g., Shen and Mickley, 2017; Yang et al., 2014, 2022). Based on the MET experiments with anthropogenic emissions kept unchanged, the changes in large-scale circulations show a weak influence on the U.S. $O_3$ trends in summer (Fig. 9a) but cause a significant $O_3$ rise in the central U.S. in winter (Fig. 9b). Averaged over the U.S., the near-surface $O_3$ mixing ratio in winter increases at a rate of 0.7±0.3 ppb/decade during 1995–2019 in MET experiments. It suggests that the variation in the large-scale circulation is responsible for 15% of the increasing trend in wintertime $O_3$ mixing ratio by 4.7±0.3 ppb/decade in the U.S. during 1995–2019 simulated in BASE experiment.
    The changes in atmospheric circulation pattern support the above finding. Compared to 1995–1999, anomalous northerly winds locate over high latitudes of North America in 2015–2019 (Fig. 9c), strengthening the prevailing northerly winds in winter. In addition, an anomalous subsidence occurs over the central U.S. in 2015–2019, compared to 1995–1999 (Fig. 9d). The anomalous subsidence transport $O_3$ from high altitudes and even stratosphere to the surface and the strengthened winds transport $O_3$ from remote regions (e.g., $O_3$ produced by Asian $NO_x$ emission) to the central U.S., both contributing to 0.2±0.1 ppb/decade of the $O_3$ increase over the U.S. (Fig. 10). The finding is consistent with Lin et al. (2015) that variations in the circulation facilitate $O_3$

transport from upper altitudes to the surface, as well as foreign contributions from Asia. The anomalous atmospheric circulation is likely linked to the location of the midlatitude jet stream, which is influenced by ENSO cycle."

8) Further, I am somewhat surprised by the statement "The mixing ratio is sometimes expressed as concentration in many studies, so we prefer to keep it as it is." Mixing ratios and concentrations are two complete different things. Please check for example this Eos article:
https://agupubs.onlinelibrary.wiley.com/doi/pdf/10.1029/00EO00007
Please correct your manuscript accordingly.

Response:
    We have now corrected the descriptions as "mixing ratio" throughout the manuscript.

Some more detailed comments:

l78: Please do not use the term 'contribution' when talking about the perturbation method.

Response:
    We revised it to "when being used to estimate the impacts of changes in multiple sources".

l107ff:As mentioned in the first review: there are methods on the global scale for sectoral attribution (e.g. Emmons et al., 2012, Grewe et al. 2017, Butler et al., 2018)

Response:
    We have now added these references.

l291ff: I don't understand this sentence. Do you mean inland shipping?

Response:
    We have revised it to "Due to a strong chemical sink associated with photolysis of $O_3$ with subsequent production of hydroxyl radical (OH) from water vapor in summer (Johnson et al., 1999), the effect of increased **international shipping emissions over the remote ocean regions** on the continental United States was blunted."

l307ff: Please reference Fig 5 b/d here

Response:
    Added the reference.

l328f: Do you mean near shore shipping emissions or the actual activity data?

Response:

    We mean near shore shipping emissions and modified this sentence as "The decrease in near-shore shipping emissions …"

l374f: Where can I see the trends 1.2 / 1.5 ppb/decade? Please explain (also in the manuscript).

Response:

    We now revised it as "due to the weakened $NO_x$ titration. Increases in aviation and shipping emissions **together** explain the 1.2±0.1 and 1.5±0.1 ppb/decade of $O_3$ trends in WUS and EUS, respectively".

396f: I can't follow the conclusion why changes in anthropogenic emissions are the main factor from what is written here. Please clarify and rephrase if needed.

Response:

    Because the trends shown by the MET experiments with the variation in large-scale circulation alone only account for a small fraction of the trends in the BASE experiments with the combined effect of large-scale circulation and emissions.
    However, considering the large-scale circulation contributes to 15% of the $O_3$ trend in winter, which is also an important factor driving the $O_3$ change, we have deleted this sentence.

l449f: What is the role of the emissions for the bias?

Response:

    We have added it as "Large uncertainties existing in the emissions also result in the biases in the $O_3$ simulation." In this study, we applied the latest CEDS version 20210205, which has corrected several biases as compared to its previous version 20160726.

l462ff: Why is the part about the bias over China deleted?

Response:

    We have added this sentence again.

l494ff : Please quantify the differences

Response:

    Quantified as "In 2014, the global total anthropogenic emission of $NO_x$ in

CEDS$_{2021}$ is about 10% lower than the CEDS$_{2016}$ estimate. This difference is mainly reflected in the NO$_x$ emissions in China and India. CEDS$_{2021}$ has a lower estimate of the global NMVOCs emission than CEDS$_{2016}$ by more than 10% during the recent decades, attributed to lower emissions from Africa, Central and South America, the Middle East and India."

l500f: The sentence about EDGAR seems misplaced here. Either include a proper comparison with EDGAR (btw. V 5.x is available) or delete this sentence.

Response:
    We have deleted it.

Reference:

Hoesly, R., O'Rourke, P., Braun, C., Feng, L., Smith, S. J., Pitkanen, T., Siebert, J., Vu, L., Presley, M., Bolt, R., Goldstein, B., and Kholod, N.: CEDS: Community Emissions Data System (Version Dec-23-2019), Zenodo, https://doi.org/10.5281/zenodo.3592073, 2019.

Cheng, J., Tong, D., Liu, Y., Yu, S., Yan, L., Zheng, B., Geng, G., He, K., and Zhang, Q.: Comparison of current and future PM2.5 air quality in China under CMIP6 and DPEC emission scenarios, Geophys. Res. Lett., 48, e2021GL093197, https://doi.org/10.1029/2021GL093197, 2021.